# What Moves the Eyes: Doubling Mechanistic Model Performance Using Deep Networks to Discover and Test Cognitive Hypotheses

**Federico D'Agostino**
Tübingen AI Center, University of Tübingen
federico.dagostino@bethgelab.org

**Lisa Schwetlick**
EPFL
lisa.schwetlick@epfl.ch

**Matthias Bethge**[*]
Tübingen AI Center, University of Tübingen
matthias@bethgelab.org

**Matthias Kümmerer**[*]
Tübingen AI Center, University of Tübingen
matthias.kuemmerer@bethgelab.org

## Abstract

Understanding how humans move their eyes to gather visual information is a central question in neuroscience, cognitive science, and vision research. While recent deep learning (DL) models achieve state-of-the-art performance in predicting human scanpaths, their underlying decision processes remain opaque. At an opposite end of the modeling spectrum, cognitively inspired mechanistic models aim to explain scanpath behavior through interpretable cognitive mechanisms but lag far behind in predictive accuracy. In this work, we bridge this gap by using a high-performing deep model—DeepGaze III—to discover and test mechanisms that improve a leading mechanistic model, SceneWalk. By identifying individual fixations where DeepGaze III succeeds and SceneWalk fails, we isolate behaviorally meaningful discrepancies and use them to motivate targeted extensions of the mechanistic framework. These include time-dependent temperature scaling, saccadic momentum and an adaptive cardinal attention bias: Simple, interpretable additions that substantially boost predictive performance. With these extensions, SceneWalk's explained variance on the MIT1003 dataset doubles from 35% to 70%, setting a new state of the art in mechanistic scanpath prediction. Our findings show how performance-optimized neural networks can serve as tools for cognitive model discovery, offering a new path toward interpretable and high-performing models of visual behavior. Our code is available at https://github.com/bethgelab/what-moves-the-eyes.

## 1 Introduction

Every time we view a scene, our eyes produce a sequence of rapid movements (saccades) and brief pauses (fixations), selectively directing our high-resolution fovea to parts of the visual world. This active sampling process—captured in so-called scanpaths—reveals a great deal about how we perceive, attend to, and make sense of our surroundings. Understanding what governs fixation selection is therefore a key question in visual neuroscience and cognitive science and also receives substantial interest from computer vision due to many applications ranging from compression [31] to design and layouting [15, 16].

Indeed, fixation selection has been extensively studied in a variety of scenarios, from reading [65, 23],

---

[*]Shared senior authorship

39th Conference on Neural Information Processing Systems (NeurIPS 2025).

to scene viewing [13, 93, 80, 45], to performing real-world tasks [33, 53, 64]. Here we focus specifically on modeling free viewing of natural scenes.

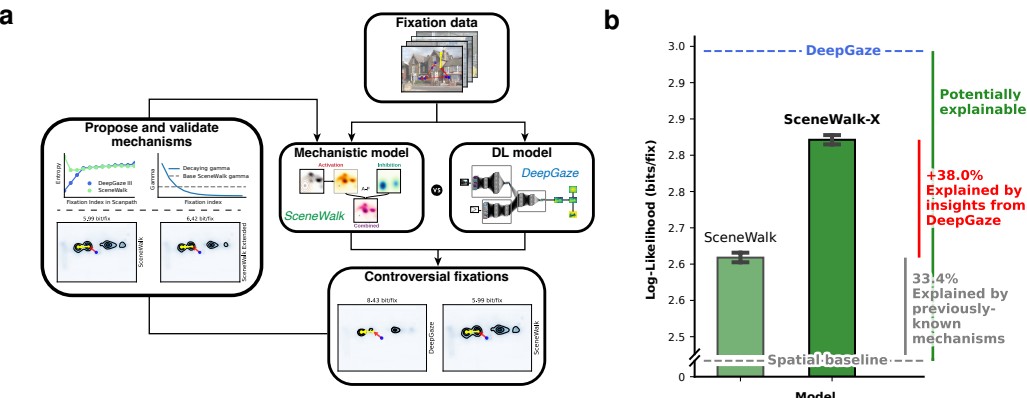

Figure 1: (a) We systematically compare prediction performances of a mechanistic scanpath model (SceneWalk) to a high-performing DNN model (DeepGaze III) to find situations that could be predicted very well but are not by the mechanistic model. Inspecting these extreme cases results in ideas for effects than can be confirmed through further analyses and give rise to mechanisms that are added to the mechanistic model. (b) This process yields three new mechanisms that double SceneWalk's predictive performance on MIT1003, substantially narrowing the explainability gap.

Approaches to this question in the modeling literature can be viewed as occupying different points on a Pareto front trading off predictive performance with direct mechanistic interpretability. At one end, models like SceneWalk [24, 68, 70] are constructed from a small set of interpretable components that directly reflect cognitive hypotheses about attentional dynamics, oculomotor constraints, and perceptual processing. These approaches align with the view of scientific understanding as a form of compression [87]: the goal is to explain complex phenomena through a small set of generalizable principles, a critical aim even when performance is not maximized. At the other end of the spectrum, performance-oriented deep neural networks (DNNs) like DeepGaze III [51] take a data-driven approach, learning to predict human fixations from large-scale data without explicitly modeling the underlying mechanisms. Their strength lies in establishing a high mark for predictive accuracy, revealing the extent of predictable structure in the data.

As common for the prediction of complex behavior, deep scanpath models indeed outperform mechanistic ones in predicting where people look [44]. But this predictive success comes at a cost: While DNNs can embed mechanisms and offer scientific insights[91, 51], they often capture complex behavioral patterns in high-dimensional parameter spaces that are not immediately transparent [27]. Conversely, mechanistic models, while more transparent, typically under-perform due to their parsimonious parametrization and due to the same strong prior assumptions that make them great testbeds for formulating and evaluating theoretical principles. This leaves a gap: We know that certain behaviors are robustly predictable by data-driven models, but we lack a compact, mechanistic account of them. One flourishing line of research addresses this by starting from the predictive end of the modeling Pareto front, developing more transparent, hybrid, or task-constrained DNNs. In this work, however, we pursue a complementary strategy embracing the mechanistic modeling philosophy: starting from the interpretable end, we ask how much of the performance gap can be closed by systematically improving the fully interpretable model.

A principled way to close this gap, then, is to focus on the mechanistic model's largest prediction errors. In the domain of static spatial saliency, model failures are often assessed by comparing model predictions to inter-observer consistency, which serves as an empirical upper bound on predictability [49, 14, 12, 45]. However, in high-dimensional distributions such as scanpaths, estimates of inter-observer consistency have been so far intractable: Each fixation is sampled from a conditional distribution $p(f_{i+1}|f_{\leq i}, I)$ that depends on the unique history of previous fixations. Because scanpaths are highly idiosyncratic, the same fixation history is almost never observed more than once, leaving us with a single empirical sample for each high-dimensional distribution. This sparse coverage makes it challenging to form stable, non-parametric estimates of inter-observer agreement.

Here is where a high-performing DNN becomes indispensable. We propose to use DeepGaze III not just as a benchmark, but as a dense, queryable approximation of the true data-generating process [41, 2]. By directly comparing the predictions of DeepGaze III and SceneWalk at the level of individual fixations, we identify *controversial fixations*: Cases where the models strongly disagree. These disagreements highlight specific behavioral patterns captured by the deep model but missed by the mechanistic one. We use them as entry points for analysis, confirming with empirical data whether the behavior is human-like, and then reformulating the missing tendencies as explicit mechanisms within the SceneWalk framework. A key question at the outset was whether the discrepancies between the models would reveal a few systematic, missing principles or a myriad of subtle, complex effects. Interestingly, as we will show, the disagreements clustered into clear, actionable categories, validating the approach. Subsequent model comparisons between versions of the mechanistic model provide a strong test for the newly added mechanisms. This iterative process repurposes the deep model as a tool for mechanistic discovery, helping quantify *how much* specific mechanisms contribute to scanpath behavior and bridging the gap between prediction and understanding. As such, our primary contribution is not the discovery of entirely novel phenomena—which would also have been possible—, but a systematic, model-guided methodology for prioritizing which of the many plausible mechanisms most improve prediction on naturalistic data.

We evaluate our method on four free-viewing datasets and demonstrate that augmenting SceneWalk with mechanisms inspired by DeepGaze leads to consistent performance improvements, bringing interpretability closer to predictive performance by halving the explanatory gap on most datasets. In doing so, we offer a roadmap for principled model development that leverages the complementary strengths of hypothesis- and data-driven approaches.

## 2 Related Work

**Scanpath Modeling**   While the field of eye movement prediction has been focused for a long time mainly on spatial fixation density prediction ("saliency prediction", [36, 11, 38, 47, 56, 22], see [7, 45] for an overview), the problem of predicting the dependencies within scanpaths of fixations has also attracted substantial interest. The seminal model of Itti and Koch [36] already predicted scanpaths, and since then a range of different mechanistic models have been proposed. Many models take direct inspiration from neuroscience and biology (MASC [1], LATEST [81], Star-FC [89], [40]) or build on statistical ideas such as CLE [5], SaccadicFlow [19], WALD-EM [43] and others [10, 79, 55, 54, 21, 90, 94], and some models implement ideas from cognitive science and attention research, such as SceneWalk [24, 68, 70], Exploration-Exploitation [59] and ROI-LSTM [78]. Finally, the progress in machine learning has brought deep-learning based models that are either only partially interpretable or complete black boxes such as SaltiNet [4], PathGAN [4], DeepGaze III [51], GazeFormer [60], ScanDMM [77] and HAT [92]. Many scanpath models take as given a spatial saliency map of the image, however, there are also models that incorporate more complex interactions of scene content and scanpath dynamics. An extensive comparison [44, 51] showed that SceneWalk is the best mechanistic model in terms of explained information, which is why we are using it as the starting point of our project. While here we focus on free-viewing behaviour, other models also take into account task dependency, such as visual search [1, 34, 91, 71], or inter-subject differences [17, 37].

**DNNs as scientific tools**   One conceptual foundation of our work is the idea that DNNs can serve as queryable empirical stand-ins for Bayesian ideal observers in domains where the analytical solution is intractable [41]. A direct extension of this idea is Scientific Regret Minimization (SRM [2]): using expressive machine learning models as powerful approximators of the data-generating process, leveraging their predictions to identify systematic failures in simpler, hypothesis-driven models. Crucially, this emphasizes cases where we know—because the DNN predicts accurately—that the behavior should have been predictable, but the mechanistic model fails. This reframes the role of DNNs not merely as predictive tools, but as guides for scientific inquiry.

**Model comparisons for insight and hypothesis generation**   Our method also relates to work comparing model predictions to extract insights or derive interpretable mechanisms. Classic approaches include some early work in model distillation, where the goal is to extract simpler, interpretable models from complex DNNs—e.g., distilling neural networks into decision trees [26]. Other work frames model comparison as a way to generate maximally informative contrasts between hypotheses

or models. An example of this is the Maximum Differentiation (MAD) Competition for perceptual models [85], or the use of controversial stimuli to pit vision models against each other in regions of disagreement [30]. These approaches resonate with our method of using disagreement between models to uncover where new mechanisms may be required.

Finally, it is important to distinguish our approach from a different line of research in *post hoc* mechanistic interpretability methods, such as feature attribution [42, 95], which aim to explain black-box models' internals. In contrast, we treat the DNN as an opaque but predictive reference, and focus instead on refining an interpretable model by learning from its disagreements with the DNN.

## 3   Methods and Experimental Setup

**Modeling Framework and Evaluation**   Our models and evaluations are formulated in the framework of next-fixation-prediction [34, 45]: Eye movements can be considered a sequential decision process in which each decision is about where to look next. This means that given an image $I$ and a sequence of previous fixations $f_0, \ldots, f_i$, models predict the next fixation $f_{i+1}$ as a conditional probability distribution $p(f_{i+1} \mid f_i, \ldots, f_0, I)$, or for short $p(f_{i+1} \mid f_{\leq i}, I)$. Compared to predicting whole scanpaths, the framework of next-fixation-prediction comes with the advantage that it allows to inspect model predictions for each individual fixation in the dataset and is compatible with classic spatial fixation density prediction: it constitutes the special case $p(f_{i+1} \mid f_{\leq i}, I) = p(f_{i+1} \mid I)$ where each fixation is predicted independently of previously made fixations. Model predictions can naturally be scored via log-likelihood $\log p(f_{i+1} \mid f_i, \ldots, f_0, I)$ which we report in bits and relative to the uniform baseline model as $\log p(f_{i+1} \mid f_i, \ldots, f_0, I) - \log p_{\text{uniform}}(f_{i+1} \mid I)$ (sometimes also termed *information gain, IG* [48]). Information gain has been shown to be a preferable metric due to its sensitive nature [50] and we use it as our main metric of predicton quality. The traditionally often used scanpath similarity metrics [3] are not applicable in our case because they don't allow single-fixation analyses and have additional conceptual problems [44].

**Explainable Information Gain**   Because information gain constitutes a ratio scale [76], differences and relative changes are meaningful. This forms the basis of *explainable information gain* [48], which quantifies how much of the theoretically possible performance a model achieves. In spatial saliency, this is measured between a lower bound (i.e. center bias) and an upper bound (i.e. gold-standard empirical model), with the difference defining the explainable information gain $\log p(f \mid I) - p(f)$. In our setting, we are interested in modeling the dependency of fixations on scanpath history. Therefore, the static gold standard from spatial saliency becomes our lower baseline $p(f_{i+1} \mid I)$. The upper bound, a principled gold standard model of $p(f_{i+1} \mid f_{\leq i}, I)$, is intractable to estimate non-parametrically from empirical data due to the sparsity of unique scanpath histories. Since each trial generates a unique fixation history, we are typically left with only a single data point for each high-dimensional conditional distribution, precluding direct estimation of inter-observer consistency. We therefore approximate this ceiling with what is to our knowledge the best performance-optimized model for scanpath prediction, DeepGaze III [51], which serves as a queryable proxy for human predictability. As a model, it also enables controlled analyses of the data distribution that sparse empirical observations alone do not permit (see Fig. 5).

**Controversial Fixations**   At the heart of our analysis technique is what we call *controversial fixations*, based on the concept of controversial stimuli [30]. A controversial fixation is a fixation for which the predictions from DeepGaze III and SceneWalk, given the previous scanpath history, differ substantially. We use two different measure to quantify these differences: First, we use the *log-likelihood difference (LLD)* $\log p_{\text{DG3}}(f_{i+1} \mid f_{\leq i}, I) - \log p_{\text{SW}}(f_{i+1} \mid f_{\leq i}, I)$. Large values mean that DeepGaze III assigns much more probability to a given fixation than SW, e.g. 8 times as much for a difference of 3 bit. Such differences suggest that while the fixation is predictable, the mechanisms currently implemented by SceneWalk do not capture it well. Additionally, we use the *weighted log-likelihood difference (WLLD)* $p_{\text{DG3}}(f_{i+1} \mid f_{\leq i}, I) (\log p_{\text{DG3}}(f_{i+1} \mid f_{\leq i}, I) - \log p_{\text{SW}}(f_{i+1} \mid f_{\leq i}, I))$. By weighting with DeepGaze's probability, we select fixations with large LLD that are additionally considered very likely by DeepGaze. We chose these fixation-centric metrics over full-distribution measures like KL-divergence because they focus the analysis on the actually observed fixation and are substantially more computationally efficient. This means that we focus more on cases that DeepGaze considers easy and which might hence showcase effects more clearly.

These controversial fixations serve as entry points for further analysis. Rather than proposing mecha-

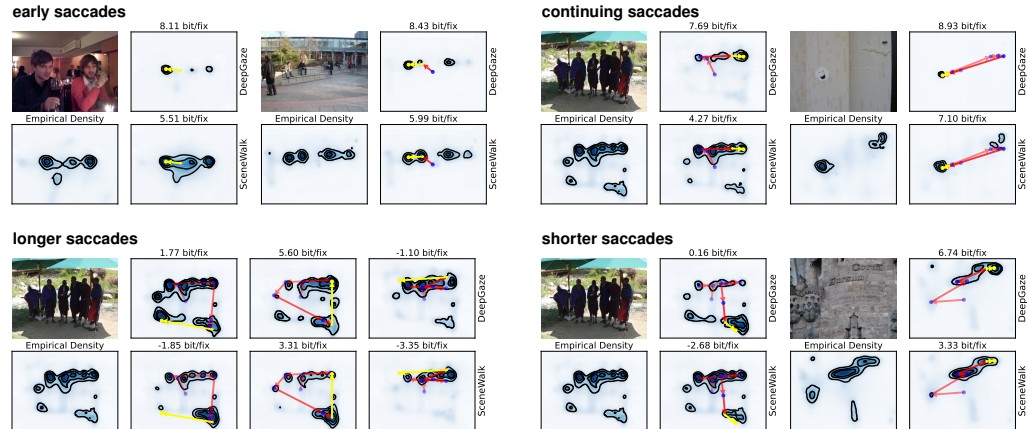

Figure 2: Controversial Fixations: Those fixations in the dataset where SceneWalk loses most performance compared to DeepGaze III in terms of LLD or WLLD. Numbers on top of predictions indicate the achieved log-likelihood relative to a uniform model for the fixation in question.

nisms *ad hoc* from single examples, we use these disagreements to guide targeted comparisons across models and empirical data. Patterns that generalize across fixations are formalized and integrated into the SceneWalk model. This provides a principled alternative to intuition-driven model development, using prediction disagreement to identify which candidate mechanisms are most impactful.

## 4 Experimental Setup

**Datasets** We use the MIT1003 dataset [38] for our controversial fixations experiments. It is well established in the gaze prediction community since it is the training set for the MIT300 benchmark dataset [39] of the MIT/Tübingen Saliency Benchmark [46]. It features a broad range of mostly photographic images, including indoor, urban, and natural scenes, which are viewed by 15 observers for 3 seconds each. All training and evaluation on this dataset employs a 10-fold cross-validation scheme across images (with separate training, validation and test splits, see Appendix D), ensuring that models are always tested on stimuli (and their associated fixations) held out during training.

We additionally use the DAEMONS [72], COCO Freeview [18] and Potsdam corpus [69] datasets for additional evaluations of the mechanisms added through our controversial fixations analyses on MIT1003.

**Spatial Baseline** Our spatial baseline model tries to predict the average spatial fixation density $p(f \mid I)$ on images, i.e. without taking scanpath dependencies into account. We require it for two reasons: Firstly we need it for putting scanpath model performances into perspective: performance gains compared to the spatial baseline show how well models exploit the scanpath dynamics. Secondly, as is the case for many scanpath models, the SceneWalk model requires as input a spatial saliency map: a map encoding which image areas are interesting independent of scanpath dynamics. Consistent with previous applications of the model [24, 70] we use our spatial baseline model to that end. Our spatial baseline model is a mixture model of a KDE of empirical fixations and predictions from a SOTA saliency model, where the parameters are fitted per image. For details see Appendix C.

**SceneWalk** The SceneWalk model [24, 68, 70] is a mechanistic scanpath model inspired by results from cognitive science. It implements an activation map and an inhibition map, which evolve over time depending on the fixated image locations. The activation map is updated for each fixation with a Gaussian window on the saliency map and decays over time, encoding what the model knows about relevant parts of the image. The inhibition map is updated for each fixation with a Gaussian at the fixation location and also decays over time, encoding which image areas the model has already explored. To predict the next saccade target, activation and inhibition maps are postprocessed with an exponential nonlinearity and combined to yield the fixation selection map. Here, we use the latest version of SceneWalk [70] which additionally includes a peri-saccadic attention shift mechanism:

Prior to a saccade, attention is already shifted towards the saccade target and hence, after the saccade, a small area in the direction of the saccade is uncovered. This mechanism results in higher prediction performance and explains the directional statistics of scanpaths better. SceneWalk has only about 20 parameters, which we fit using MAP estimation (for details see Appendix E).

**DeepGaze III**  DeepGaze III [51] is a deep learning based scanpath model. For predicting the next fixation in a scanpath, it takes as input the viewed image and the last four fixation locations. The input image is encoded with a pretrained backbone and processed by a priority network that decodes a spatial priority map. This spatial priority map is then combined with the information about previous fixation locations in a fixation selection network to output a priority map that is conditioned on the previous scanpath history. This map is blurred, combined with a center bias and passed through a softmax to yield the probability distribution for the upcoming fixation location. For better comparability with SceneWalk, here we removed the priority network and instead used the same empirical densities that are also used by SceneWalk in the fixation selection network. This serves to make sure that all differences between the two models are necessarily due to how they combine the spatial priority map with the fixation history and avoids confounders. We fit DeepGaze III in this setting on the MIT1003 dataset using 10-fold crossvalidation across images (see Appendix E).

## 5  Experiments and Results

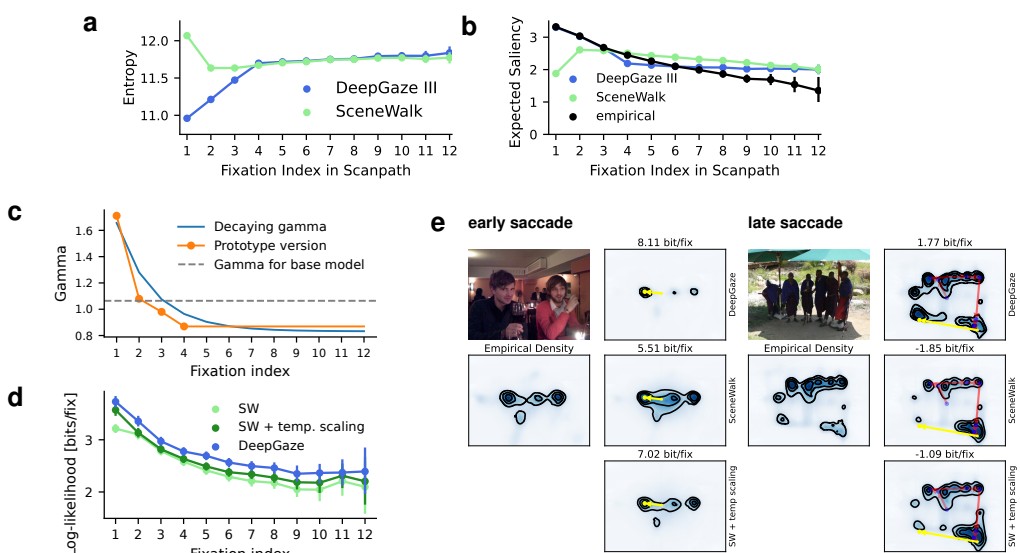

Figure 3: Time dependent temperature scaling. We found that DeepGaze shows higher confidence (less entropy) early on (a) and predicts fixations in more salient locations (b) in agreement with the empirical data. SceneWalk is not showing this effect. To address this, we introduced fixation-index-dependent temperature scaling (c), modeled with exponential decay (d), which improves predictions for both early and late fixations (e).

### 5.1  Model Comparison

**Explainable information**  By evaluating our spatial baseline, DeepGaze and SceneWalk on the held-out test cross-validation splits of the MIT1003 dataset, we find that DeepGaze explains 0.43 bit/fix more than the spatial baseline. From estimate of the explainable information gain , the SceneWalk model explains with 0.15 about one third, leaving an explanatory gap of two thirds.

**Controversial Fixations**  After the full-dataset level analysis, we now go to the level of individual fixations and inspect those fixations where SceneWalk shows the largest failures. For both log-likelihood difference and weighted log-likelihood difference, we select the top six images with largest mean score and from them visualize the six fixations with highest individual scores. We find that

nearly all (92%, see Appendix J) of these controversial fixations can be classified into four categories (Figure 2): DeepGaze III seems to gain by predicting that (1) early saccades go exclusively to the most salient targets and that (2) often saccades tend to continue in the direction of the previous saccade. In addition, DeepGaze gains by having a better understanding of when saccades are (3) longer or (4) shorter.

## 5.2 New Mechanisms in SceneWalk

**Time Dependent Temperature Scaling** Controversial fixations revealed that DeepGaze favors high-saliency targets early in scanpaths, with this effect diminishing over time (Fig. 2). We confirmed that DeepGaze indeed shows higher confidence (less entropy) early on (Fig. 3a) and predicts fixations in more salient locations in agreement with the empirical data (Fig. 3b). SceneWalk partially captures this effect thanks to the buildup of the inhibitory stream over time which forces the model to increasingly attend to less salient locations. However, the effect in SceneWalk is clearly weaker than in DeepGaze and very off for the first fixation. The original SceneWalk model already uses exponents to shape the activation and inhibition maps to be more or less deterministic, i.e. emphasize highly activated areas. This essentially regulates the temperature of the distributions. To model the change in the focus on salient locations, we now made this exponent in SceneWalk dependent on the fixation index in the scanpath (Fig. 3c). We first used different independent exponents for earlier and later fixations ("prototype version" in Fig. 3c). These exponents closely resembled an exponential decay and hence for the final version we fitted the exponent as an exponential decay over the course of a scanpath (details in Appendix B.1). The result is a strong early preference for salient locations, with the exponent decaying below the baseline model's value around the third fixation, allowing more exploratory behavior later in viewing. This improves prediction performance both for early and late fixations (Fig. 3d) and also visibly improves the behavior on the corresponding controversial fixations (Fig. 3e). Interestingly, it has been hypothesized before that the temperature of the fixation distribution might change over presentation time [69].

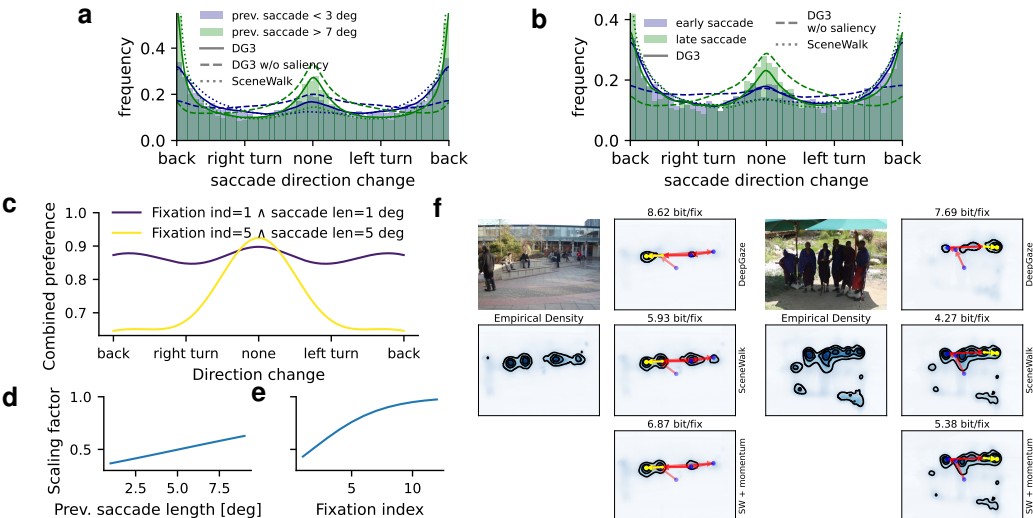

Figure 4: Saccadic momentum: Controversial fixations suggested that DeepGaze sometimes prefers saccade to continue in the same direction. We confirmed such a *saccadic momentum* effect especially after long saccades (a) and later in scanpaths (b). These effects remain even after controlling for the distribution of salient objects (dashed lines), indicating a genuine directional bias. We added a saccadic momentum and return mechanism to SceneWalk (c) modulated by previous saccade length (d) and fixation index (e), improving predictions for "ongoing saccades" controversial fixations (f).

**Saccadic Momentum** A second prominent cluster of controversial fixations clearly indicated that DeepGaze more accurately predicted short, continuing fixations mostly following long saccades. We found the effect to be consistent with known phenomena of saccadic momentum [74, 75, 88], which have also been previously used in scanpath models [40]. We noticed that this behavior is

particularly pronounced after long saccades [75] and during later stages of the scanpath (Fig.4a,b). While some of this structure could be explained by saliency alone, control analyses using DeepGaze evaluated on uniform empirical fixation densities (i.e., with saliency effectively removed) revealed that the momentum effect largely persists (Fig.4d, dashed lines), suggesting it reflects some form of oculomotor bias rather than image content. Besides momentum, both DeepGaze and the empirical data exhibited a balance between returning to previously attended regions and continuing in the same direction, especially during early viewing and following short saccades.

SceneWalk's perisaccadic attention shift can partially explain such behavior, but much more weakly than in the data (see Appendix Fig 7 for a larger figure). To address this, we introduced a dynamic oculomotor bias map that captures both saccadic momentum and return tendencies. The mechanism modulates directional preferences based on the length of the previous saccade and the position within the scanpath and applies the resulting bias map to SceneWalk's internal probability map (details in Appendix B.2). Following short saccades early in a scanpath, the model learns to balance return saccades and momentum, while after long saccades and later in viewing, it favors continued movement in the same direction (Fig. 4c). This extension significantly improves SceneWalk's ability to capture directional patterns in controversial fixations, bringing its predictions closer to those of DeepGaze (Fig. 4f). Again, this mechanism aligns with prior literature suggesting that saccadic momentum and return effects may jointly reflect oculomotor constraints or facilitation-of-return mechanisms [74, 88, 67], and even more complex dynamics beyond muscle physics [62].

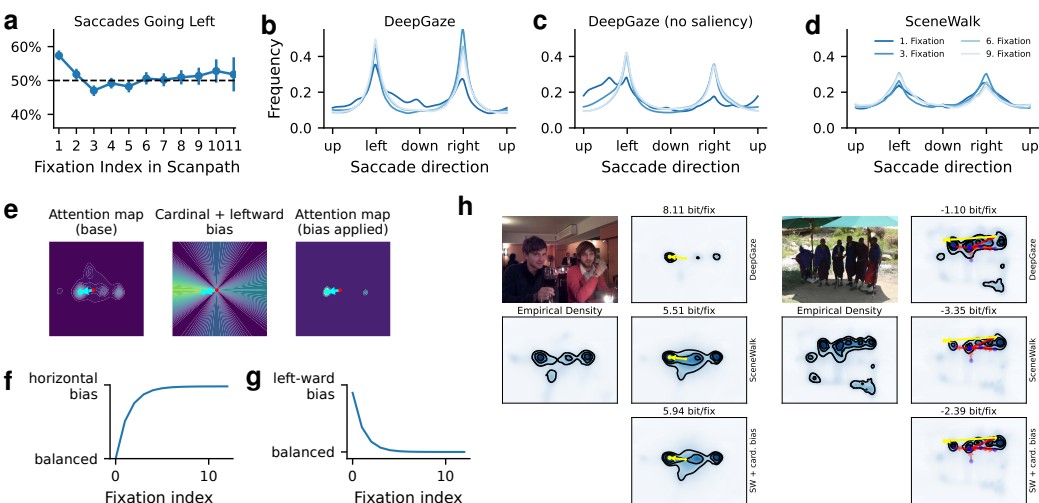

Figure 5: Horizontal and left-wards attention bias: Controversial fixations suggested that DeepGaze predicts early saccades to go to the left. This is indeed the case in the data (a) and the DeepGaze predictions (b). The effect persists when DeepGaze is run with uniform saliency maps (c) while SceneWalk (d) shows a too weak effect that increases over time. Therefore, we added a cardinal attention bias to SceneWalk, than can additionally have a left-asymmetry (e) and can adapt over time (f, g). This improves predictions on the relevant controversial fixations (f).

**A horizontal and left-wards attentional bias** Especially for early controversial fixations, DeepGaze seemed very confident about left-wards saccades (Figure 2). We found his effect indeed present in the data (Fig. 5a), as previously reported elsewhere [25]. We tested how well the models reflected the effect by computing, for each individual fixation, a histogram of expected saccade directions and averaged them for earlier and later fixations. This confirmed that DeepGaze learned a left-wards and more generally a horizontal bias (Fig. 5b). Importantly, the bias is still there when we evaluate DeepGaze with uniform saliency maps (Fig. 5c), confirming that the effect is not not purely saliency-driven. Interestingly, DeepGaze's leftward bias fades over time, while its horizontal bias builds up across early fixations. SceneWalk, in contrast, shows only weak directionality and an incorrect increase in leftward bias over time (Fig. 5d), possibly reflecting artifacts from the empirical densities.

We decided to account for these directional biases by applying a variable cardinal prior to SceneWalk's attention map before saccade selection. This modification is motivated by two complementary observations. First, a purely oculomotor bias—like the one implemented by [40]— is unlikely to explain the found preference for early saccades to the left or the weaker horizontal bias for early fixations even in the absence of saliency. Second, beyond known effects of cardinal oculomotor biases, empirical evidence suggests that attentional processes themselves exhibit anisotropies across the cardinal axes—particularly favoring horizontal over vertical directions [83, 57, 82]. Notably, attentional modulation has been observed as early as the lateral geniculate nucleus [57] lending biological plausibility to an attentional bias operating in concert with oculomotor constraints.

Concretely, our cardinal attention bias is implemented as a smooth function of saccade angle, with preference peaking along the cardinal axes and decaying with distance from the current fixation. Additionally, we model a dynamic asymmetry between horizontal and vertical directions, with the horizontal bias gradually increasing over the course of a trial (Fig. 5f). Finally, to account for the early leftward bias, we also introduced an asymmetry that favors leftward over rightward saccades during early fixations, with strength that decays over time (Fig. 5g). The resulting cardinal attention bias map is combined multiplicatively with the attention map (Fig. 5e). For details see Appendix B.3.

### 5.3 Results

**A new SOTA for mechanistic scanpath prediction**  On the MIT1003 dataset, our full extended SceneWalk model (which we term SceneWalk-X for short), doubles the performance of the original SceneWalk perisaccadic model compared to the spatial baseline. It further closes $56\%$ of the remaining gap in explainable performance as estimated by DeepGaze. Since SceneWalk was the previous SOTA on MIT1003 in mechanistic scanpath prediction [51], this sets a new SOTA in mechanistic scanpath prediction. Notably, we evaluated SceneWalk-X also on the DAEMONS dataset [72], which is specifically designed for scanpath prediction and includes longer scanpaths on high-resolution natural scenes. SceneWalk performs better when trained and evaluated on DAEMONS, where it already explains more than $60\%$ of the explainable information relative to the baseline. SceneWalk-X closes $40\%$ of the residual gap in explainable performance between DeepGaze and the base SceneWalk model. While this is smaller than the gains observed on MIT1003, it is worth noting that DAEMONS contains longer, more complex scanpaths where the spatial baseline alone is less predictive, and the original SceneWalk model already performs well—consistent with its design focus on extended viewing behavior. Nonetheless, SceneWalk-X still nearly halves the remaining gap to DeepGaze, demonstrating its robustness and leaving room for future improvements specific to this richer dataset.

**More evaluations**  In Appendix A we confirm our results hold on two more datasets (COCO Freeview [18] and the Potsdam Scene Viewing Corpus [69]) and compare with other mechanistic models. We further report AUC scores for all datasets to show that our results do not depend on the chosen evaluation metric.

**Relevance Assessment**  To quantitativly assess the overall relevance of each added effect, we evaluated different extensions of SceneWalk and also compared to the earlier "classic" SceneWalk [24, 68] without perisaccadic attention shifts (Fig. 6). We find that on MIT1003, each of the three mechanisms contributes about equally to the overall increase in explanatory performance. On the DAEMONS dataset, the performance gain is mostly due to momentum and cardinal biases, while temperature scaling contributes little. We hypothesize that this might be because DAEMONS features much longer scanpaths, and the temperature stays mostly constant after the first few fixations.

**Reconciling controversial fixations**  In Appendix J, we show all examined controversial fixations together with the predictions from the fully extended SceneWalk model and find that many of them are now predicted substantially better.

## 6  Discussion

Data-driven (DNN) and hypothesis-driven (mechanistic) approaches to modelling cognitive and perceptual processes are usually painted as being in contrast with each other. One aims primarily to predict, the other primarily to understand and confirm hypotheses. In this work we demonstrate how

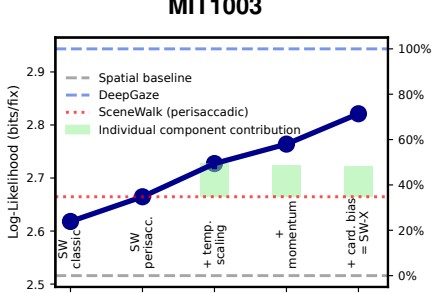
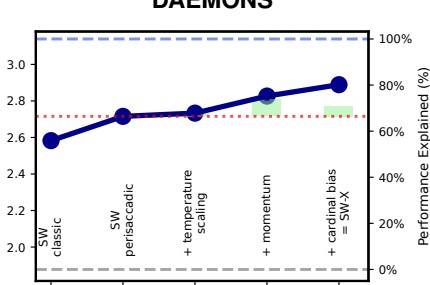

Figure 6: Contributions of the different added mechanisms on the MIT1003 and DAEMONS datasets.

both techniques can be used in concert to improve our understanding of the mechanisms underlying fixation selection. By focusing on fixations where the predictions of a deep model (DeepGaze III) and a mechanistic model (SceneWalk) diverged, we isolated specific mechanisms implicitly captured by the deep model but absent in SceneWalk. These mechanisms were then formalized and incrementally implemented into SceneWalk. Each addition was tested via its impact on predictive likelihood, thereby providing a strong test of its behavioral relevance.

The conceptual cornerstone of our method is to use a DNN model as a stand-in for predictability. This turns a black-box model into a scientific instrument: not something we trust blindly, but something we can probe, compare against, and learn from. This shift circumvents many of what are portrayed as the classic weaknesses of deep learning and focuses instead on the strenghts of such models. For example, DeepGaze III allows us to test whether certain spatial or temporal biases affect behavior beyond what is already explained by bottom-up saliency—a manipulation that would be extremely difficult, if not impossible, to achieve with empirical data alone. In this sense, deep models can offer novel ways to look at the underlying data distribution. When combined with direct comparisons to mechanistic models, this enables to more flexibly look at data and formulate hypotheses.

Importantly, the main contribution of our method is not necessarily in discovering novel mechanisms. Many of the effects we identify, have been observed before, and we mainly add nuance to the way they might be operating, which will require future work to confirm. Rather, our main contribution here lies in offering a principled procedure for identifying which among the many plausible mechanisms are most useful for improving behavioral predictions. By translating prediction errors into candidate explanations and rigorously testing them in a mechanistic framework, our method strengthens the link between predictive performance and theoretical insight.

**Limitations and outlook**     A few potential limitations and subtleties of our approach warrant further discussion. First, our method assumes that DeepGaze provides a reasonable proxy for human fixation behavior, but, like any DNN, it may learn dataset-specific biases or shortcuts [28, 27]. Second, we acknowledge that adding a mechanism which improves the mechanistic model's fit does not necessarily imply that the same mechanism is used by humans, as it could be a correlate or computational proxy. While mechanistic models aim for interpretability, they are still approximations and subject to their own inductive biases. Grounding mechanisms in empirical findings helps address this limitation. Finally, our current pipeline relies on manual identification and implementation of mechanisms, a deliberate choice to validate the methodology in a transparent, end-to-end fashion. We view this work as a foundational proof of concept. Automating this search over candidate model extensions, especially for more complex interactions across time and context, remains an open and exciting challenge [29] that points toward a generalizable, semi-automated modeling pipeline—a step towards what could be seen as an "automated scientist" [e.g., 58] for behavioral modeling. Similarly, our results are based on scanpaths over natural images; applying this framework to other domains such as reading, video, or real-world interactions is an exciting direction for future research.

**Broader impact**     Accurate scanpath models have practical relevance in fields such as visual design, education, and assistive technology. However, there are ethical considerations: predictive models of attention could be used to manipulate user behavior, for example in advertising or UX design. It is therefore important that such models are developed transparently and deployed with caution.

## Acknowledgments

The authors thank the International Max Planck Research School for Intelligent Systems (IMPRS-IS) for supporting Federico D'Agostino. This work was supported by the German Research Foundation (DFG): SFB 1233, Robust Vision: Inference Principles and Neural Mechanisms, project number: 276693517.

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

# A Comparative evaluation on more models and datasets

We evaluated our models on four datasets: MIT1003 [38], DAEMONS [72], COCO Freeview [18] and the Potsdam Corpus [69]. On all four datasets, we construct a common spatial baseline model and then train DeepGaze as well as the different SceneWalk variants using the spatial baseline model. For better comparison, we also evaluate the Constrained Levy Exploration model [CLE; 6] and the model of Kadner et al., 2023 [40]. For all models we evaluate information gain relative to the uniform baseline (LL) and AUC [44]. We find (Table 1) that on all datasets, SceneWalk-X closes a substantial part of the performance gap between the perisaccadic SceneWalk and DeepGaze, both in LL and in AUC.

As a note, temperature scaling has only a very small effect in AUC: this is to be expected, as AUC is invariant to temperature changes. The marginal score increase is likely an indirect effect, as having temperature as a free parameter during training gives the optimizer more flexibility, allowing the other model parameters to converge to a slightly better solution.

| Model | MIT1003 | | DAEMONS | | COCO Freeview | | Potsdam Corpus | |
|---|---|---|---|---|---|---|---|---|
| | LL | AUC | LL | AUC | LL | AUC | LL | AUC |
| Spatial Baseline | 2.52 | 0.9223 | 1.88 | 0.8743 | 2.08 | 0.9034 | 1.81 | 0.8768 |
| Kadner et al. 2023 [40] | - | 0.9237 | - | 0.8806 | - | 0.9113 | - | - |
| CLE (Boccignone et al. 2004, [6]) | 2.18 | 0.9240 | 2.42 | 0.9118 | 2.34 | 0.9212 | - | - |
| SceneWalk (perisaccadic) | 2.66 | 0.9304 | 2.71 | 0.9207 | 2.53 | 0.9240 | 2.32 | 0.9129 |
| + temp. scaling | 2.72 | 0.9305 | 2.73 | 0.9209 | 2.54 | 0.9241 | 2.33 | 0.9130 |
| + momentum | 2.77 | 0.9321 | 2.82 | 0.9235 | 2.58 | 0.9254 | 2.38 | 0.9147 |
| + card. bias = SceneWalk-X | 2.82 | 0.9343 | 2.88 | 0.9267 | 2.64 | 0.9275 | 2.42 | 0.9164 |
| DeepGaze | 2.94 | 0.9386 | 3.18 | 0.9365 | 2.87 | 0.9343 | 2.54 | 0.9194 |

Table 1: Performance of our models and other previous scanpath models on MIT1003, DAEMONS, COCO-Freeview and the Potsdam Scene Viewing Corpus

# B Implementation details of the new mechanisms

## B.1 Time dependent temperature scaling

To capture the observation that early fixations tend to be more strongly biased toward salient image regions, we introduce a time-dependent temperature scaling mechanism into the SceneWalk model. This mechanism dynamically adjusts the exponents used to shape the attention and inhibition maps during scanpath generation based on the fixation index. The original SceneWalk model already applies a fixed exponent to these maps to regulate their sharpness, acting similarly to a softmax temperature. Our extension generalizes this by making the exponents decay over time, allowing for sharper attention selection early in the trial and broader selection later.

Concretely, in the original SceneWalk model, the priority map $u_{ij}(t)$ at each location is computed as a subtractive combination of normalized, exponentiated attention $A_{ij}(t)$ and inhibition $F_{ij}(t)$ maps:

$$u_{ij}(t) = \frac{(A_{ij}(t))^\gamma}{\sum_{kl}(A_{kl}(t))^\gamma} - C_F \cdot \frac{(F_{ij}(t))^\gamma}{\sum_{kl}(F_{kl}(t))^\gamma}. \tag{1}$$

Here, $C_F$ is a weight parameter for inhibition, and, importantly, $\gamma$ controls the sharpness of both the attention and inhibition distributions and was constant across fixations.

We replace the constant $\gamma$ with a fixation-index-dependent value. Specifically, the exponent $\gamma_n$ for the $n$-th fixation is given by an exponential decay from an initial high value $\gamma$ toward a base value $\gamma_{\text{base}}$:

$$\gamma_n = \gamma_{\text{base}} + (\gamma - \gamma_{\text{base}}) \cdot e^{-\lambda_\gamma \cdot (n-1)}. \tag{2}$$

Note that here the fixation index in the scanpath $n$ follows a 1-based indexing.

The new trainable parameters we introduce with this formulation are:

1. $\gamma_{\text{base}}$: minimum exponent for inhibition
2. $\lambda_\gamma$: decay rate for $\gamma$

This enables the model to express a strong preference for high-saliency targets early on (large $\gamma_n$) and gradually reduce this preference (lower $\gamma_n$) over time, converging to $\gamma_{\text{base}}$.

## B.2  Saccadic momentum

To better capture directional tendencies that DeepGaze can capture in human scanpaths—specifically the tendency to continue in the same direction (momentum) and to return to previously attended regions (return saccades)—we augment the SceneWalk model with a dynamic oculomotor bias map $M_{ij}^{\text{dir}}$. The map is computed over all possible gaze targets on each fixation, and combines two angular Gaussian components: A momentum component favoring forward continuation and a return component favoring 180° reversals. These components are blended using a dynamic weighting function, which depends both on the current fixation index and the length of the preceding saccade.

Specifically, let

- $n$ be the current fixation index (with 1-based indexing),
- $s$ be the length of the previous saccade, in degrees of visual angle,
- $\theta_{ij}$ be the angular difference between the current fixation and candidate saccade directions, normalized between $-\pi$ and $\pi$.

Then we define the fixation index weight as:

$$w_{\text{fix}}(n) = \frac{1}{1 + \exp\left(-\frac{n - \mu_{\text{fix}}}{\sigma_{\text{fix}}}\right)}, \tag{3}$$

and the saccade length weight as:

$$w_{\text{len}}(s) = \frac{1}{1 + \exp\left(-\frac{s - s_{\text{short}}}{s_{\text{scale}}}\right)}. \tag{4}$$

The fixation index weight and the saccade length weight are then multiplied to get the combined direction weight:

$$w_{\text{dir}} = w_{\text{fix}} \cdot w_{\text{len}}. \tag{5}$$

We can compute the angular preference fields as:

$$G_{\text{momentum}}(\theta_{ij}) = \exp\left(-\frac{\theta_{ij}^2}{2\sigma_{\text{mom}}^2}\right),$$

$$G_{\text{return}}(\theta_{ij}) = \exp\left(-\frac{(\theta_{ij} - \pi)^2}{2\sigma_{\text{return}}^2}\right) + \exp\left(-\frac{(\theta_{ij} + \pi)^2}{2\sigma_{\text{return}}^2}\right). \tag{6}$$

These are then combined as follows, using the combined direction weight:

$$M_{ij}^{\text{dir}} = (1 - w_{\text{dir}}) \cdot G_{\text{return}}(\theta_{ij}) + w_{\text{dir}} \cdot G_{\text{momentum}}(\theta_{ij}). \tag{7}$$

Finally, the map is combined with SceneWalk's priority map $u_{ij}^{\text{final}}$ before saccade selection:

$$u_{ij}^{\text{final}} = \frac{u_{ij}^{\text{final}} \cdot (\omega_{\text{bias}} \cdot M_{ij}^{\text{dir}})}{Z}. \tag{8}$$

The new trainable parameters we introduce with this mechanism are:

1. $\sigma_{\text{mom}}$: width of the angular Gaussian for saccadic momentum (same-direction preference)
2. $\sigma_{\text{return}}$: width of the angular Gaussian for return saccades (opposite-direction preference)
3. $\mu_{\text{fix}}$: fixation index midpoint for the sigmoid controlling the momentum-return tradeoff
4. $\sigma_{\text{fix}}$: steepness of the sigmoid over fixation index
5. $s_{\text{short}}$: midpoint of the sigmoid controlling sensitivity to saccade length
6. $s_{\text{scale}}$: scale parameter of the sigmoid over saccade length
7. $\omega_{\text{bias}}$: scaling factor controlling the influence of the directional prior in the final priority map.

## B.3 Cardinal attention bias

Following comparisons with DeepGaze and consulting the literature, to capture anisotropies in visual attention and oculomotor behavior we introduce a mechanism that biases SceneWalk's attention map toward cardinal (horizontal and vertical) saccade directions. Additionally, we model a leftward bias for early fixations. These components are combined into a single potential map that modulates the attention map before it is normalized and combined with inhibition. As for saccadic momentum and temperature, the mechanism is dependent on fixation index.

Let:

- $n$ be the current fixation index (with 1-based indexing),
- $\theta_{ij}$ be the angular difference between the current fixation and candidate saccade target,
- $d_{ij}$ be the Euclidean distance of a candidate saccade target to the current fixation.

Then we formulate the cardinal bias as:

$$Q_{ij}^{\text{cardinal}} = \cos(4\theta_{ij}) \cdot \exp\left(-\frac{d_{ij}}{\tau}\right) , \tag{9}$$

where $\tau \propto \frac{1}{\chi}$ is a decay constant.

We further define the vertical/horizontal strength modulation as a scalar that increases over fixations:

$$\alpha_{\text{dir}}(n) = 1 + (\alpha_{\text{vert}} - 1) \cdot (1 - e^{-\rho n}) , \tag{10}$$

and the leftward bias as:

$$\alpha_{\text{left}}(n) = 1 + (\alpha_{\text{left}}^0 - 1) \cdot e^{-\lambda(n-1)} . \tag{11}$$

The biases are then applied as follows. First, for the horizontal/vertical bias, let $m_{ij}^{\text{vert}} = 1$ if location $(i, j)$ lies in the vertical direction relative to the current fixation (i.e., $|dx| < |dy|$), otherwise $m_{ij}^{\text{vert}} = 0$. The vertical bias is applied as:

$$Q_{ij}^{\text{cardinal}} \leftarrow Q_{ij}^{\text{cardinal}} \cdot \left[ m_{ij}^{\text{vert}} \cdot \alpha_{\text{dir}}(n) + (1 - m_{ij}^{\text{vert}}) \cdot 1 \right] . \tag{12}$$

Similarly, for the leftward bias, we define binary masks for horizontal directions:

- $m_{ij}^{\text{left}} = 1$ if $dx < 0$ and $|dx| > |dy|$, 0 otherwise;
- $m_{ij}^{\text{right}} = 1$ if $dx > 0$ and $|dx| > |dy|$, 0 otherwise;

and then apply:

$$Q_{ij}^{\text{cardinal}} \leftarrow Q_{ij}^{\text{cardinal}} \cdot \left[ m_{ij}^{\text{left}} \cdot \alpha_{\text{left}}(n) + m_{ij}^{\text{right}} \cdot \frac{1}{\alpha_{\text{left}}(n)} + (1 - m_{ij}^{\text{left}} - m_{ij}^{\text{right}}) \cdot 1 \right] . \tag{13}$$

Finally, the resulting potential map is normalized to sum to 1, and applied multiplicatively to the attention map:

$$A_{ij}^{\text{mod}} = \frac{A_{ij} \cdot (1 + \omega_{\text{mix}} \cdot Q_{ij}^{\text{cardinal}})}{Z} \tag{14}$$

The new trainable parameters we introduce with this mechanism are:

1. $\chi$: controls the spatial decay rate of the cardinal potential with distance
2. $\alpha_{\text{vert}}$: maximum vertical vs horizontal bias ratio
3. $\rho$: growth rate of vertical/horizontal anisotropy over fixations
4. $\alpha_{\text{left}}^0$: initial strength of the leftward bias
5. $\lambda$: decay rate of the leftward bias over fixations
6. $\omega_{\text{mix}}$: global scaling factor for the oculomotor potential's influence on the attention map.

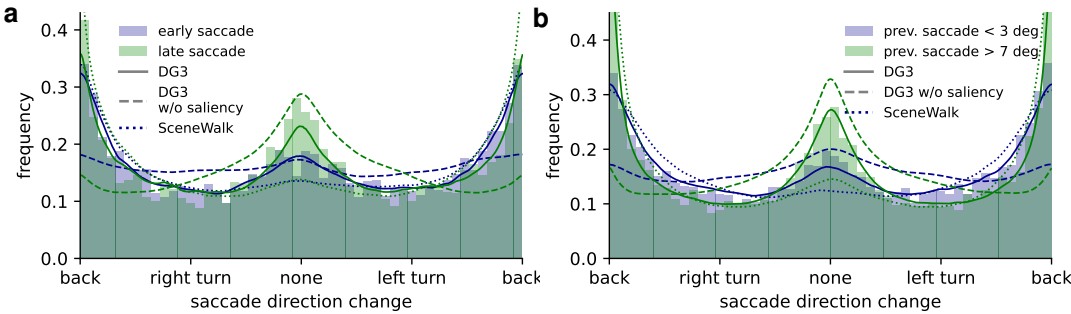

Figure 7: relative saccade direction distribution as predicted by DeepGaze, DeepGaze without saliency maps and SceneWalk for early and late saccades (a) as well as after short or long previous saccades (b).

## C Spatial baseline

Commonly, empirical fixation densities are estimated with Gaussian kernel density estimates. However, recent progress in spatial saliency modeling [52] has produced models which predict fixations from new observers better than the empirical density. While this feels counterintuitive, it is possible because saliency datasets typically have data from only 10–20 subjects per image [18, 8]). Suboptimal saliency maps could lead to confounders in our study, e.g., models might use previous fixation locations simply for estimating a better saliency map. Hence we extend the approach by [52] and estimate the average fixation density for each subject on an image as a mixture of four distributions: first, a KDE over the fixations from all other subjects, secondly the centerbias, i.e. a KDE over the fixations from all other images, thirdly a uniform component, and in addition as fourth component we add the predicted fixation density from the state-of-the-art saliency model [52]. The last component guarantees that the spatial baseline model is at least as good as the used saliency model. For a discussion of the other three components, see [45].

The parameters of the model, i.e. bandwidth and mixture weights, are optimized individually on each image for maximum likelihood. This results in a prediction for each subject, using only other subject's fixations. To compute the final fixation density for an image, we average over the predicted fixation densities for all subjects with weights that are location dependent: for each true fixation location, only the components not using this fixation in their KDE are switched on. This defines weighting coefficients for all fixated pixels such that the prediction for that pixel only comes from subjects which didn't fixate the pixel. For all other pixels, the weights are interpolated linearly, and the resulting spatial average is re-normalized. This approach removes visual artifacts due to single, potentially random fixations and gives more interpretable results.

## D Details about the datasets

The MIT1003 dataset is accessed via the pysaliency python library `https://github.com/matthias-k/pysaliency`. DAEMONS and COCO Freeview come with an official training/validation split which we are using. MIT1003 and the Potsdam Corpus do not have official validation splits. Instead, we use 10-fold crossvalidation using `pysaliency.filter_datasets.train,validation,test_fold(stimuli, fixations, crossval_folds=10, test_folds=1, val_folds=1)`. On MIT1003, the validation split has been used for tuning model hyperparameters and learning rate schedules, which have then also been used on the other datasets. All reported evaluations use the test split on MIT1003 and the Potsdam Corpus, and the official validation split for DAEMONS and COCO Freeview.

## E Details about models and training

**DeepGaze** DeepGaze III was originally trained using a four stage training paradigm including spatial pretraining on a different dataset, spatial training on the target dataset, scanpath training on

the target dataset and a final joint training phase. This procedure was necessary for not overfitting the spatial priority network while training the scanpath parts. Since we removed the spatial priority network, we only need the scanpath training phase. Since DeepGaze III was originally trained on MIT1003 dataset, which we are also using, we can apply exactly the same hyperparameters: DeepGaze is trained using the Adam optimizer with a batchsize of 4 and an initial learning rate of 0.001. In addition, we use a learning rate schedule consisting of decays of the learning rate by a factor of 10 after 10, 20, 30 and 31 epochs. After the 32nd epoch, training is stopped. Consistent with the original training, we also use 10-fold crossvalidation. For all results in this paper use for each image the model weights which have not seen this image in training.

**SceneWalk**    Originally, SceneWalk was trained fully Bayesian using MCMC. Here, we instead re-implemented the model in Jax and train it using MAP estimation, resulting in substantially faster fitting times. SceneWalk is trained via gradient descent with the Adam optimizer on a MAP objective. Parameter prior ranges are chosen to be broad, and taken mostly from Schwetlick et al. [70]. The hyperparameter configuration for the base perisaccadic model is also taken from Schwetlick et al. [70]. We use a "OneCycle" learning rate schedule [73] over 30 epochs for the MIT1003 dataset, and 50 epochs for DAEMONS, with an initial learning rate 0.001, a peak learning rate of 0.01 and a batch size of 32.

## F    Error bars

All reported error bars are bootstraped 95% confidence intervals for the mean log-likelihood per image using the normalization method of Cousineau [20] for paired comparisons with the correction of Morey [61].

## G    Assets

DeepGaze is implemented in python using `pytorch` [63] using the official implementation from `github.com/matthias-k/DeepGaze`). SceneWalk was reimplemented in `jax` [9] (Apache 2 license) based on the official implementation from `github.com/lschwetlick/SceneWalk_Model`. Model evaluations and saliency metrics were using the public `pysaliency` toolbox (`github.com/matthias-k/pysaliency`, MIT license). Also used were `scipy` [84] and `numpy` [32] for computations, `pandas` [66] for statistics and data handling as well as `matplotlib` [35] and `seaborn` [86] for plotting.

## H    Code Availability

The SceneWalk-X implementation in `jax`, along with code to reproduce our main findings can be found at `https://github.com/bethgelab/what-moves-the-eyes`.

## I    Compute Resources

All main experiments were conducted on 2080Ti GPUs. Training the DeepGaze model took about 3 days for all 10 splits and was done four times until the final model setup evolved. Training SceneWalk took about 30 minutes, while developing the mechanisms we trained about 50 model iterations. On DAEMONS, we used A100 GPUs and DeepGaze took about 4 days to train, and each SceneWalk model about two hours. Overall, we used about 18 days of compute.

## J    Controversial Fixations

Here we show the full set of controversial fixations that we inspected and classified as basis for our mechanistic extensions. We examined a total of 72 fixations, defined as the top 6 fixations from the top 6 images in both conditions (LLD and WLLD). For all controversial fixations, we show predictions from DeepGaze, SceneWalk and all mechanistic extensions of SceneWalk, including the fully extended SceneWalk-X model.

Of the four categories we detail in Figure 2, we found:

- Long saccades: 17 clear, 3 plausible

- Short saccades: 8 clear, 1 plausible

- Continuing: 13 clear, 2 plausible

- Early fixations: 30 clear

66 out of 72 fixations (92%) could be clearly assigned to one of the four categories, and the remaining 6 (8%) still plausibly belonged to a known type. We see this as a strong indicator that the analysis indeed revealed failure cases that are not idiosyncratic, but rather fall into meaningful, repeatable patterns that are interpretable and mechanistically actionable.

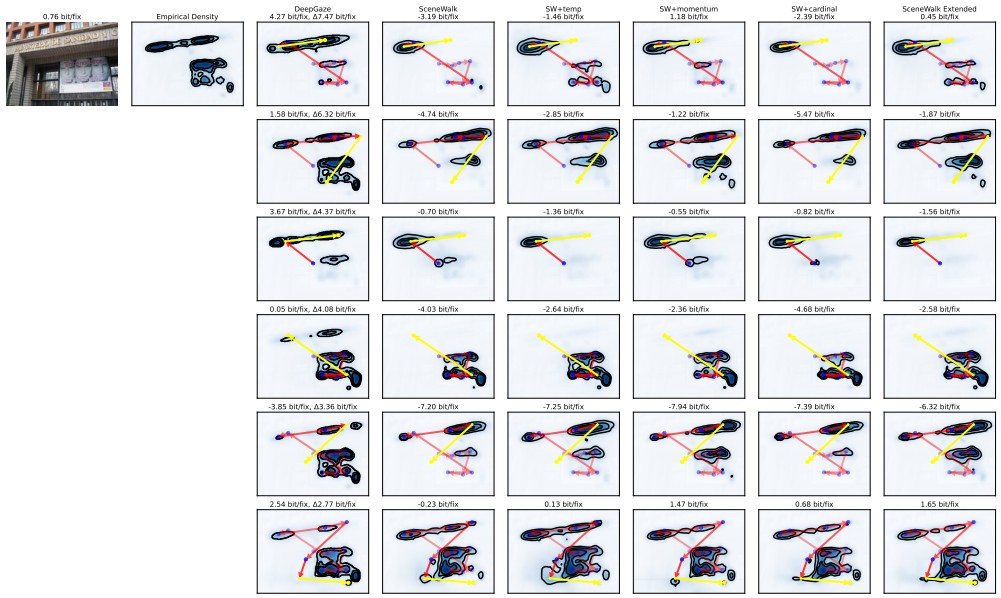

Figure 8: Controversial Fixations selected for maximum difference in log-likelihood between DeepGaze and SceneWalk, Part 1. We first select the top images with mean log-likelihood difference and then show for each image the 6 fixations with maximum log-likelihood difference. We visualize the image, the empirical densities and the predictions of DeepGaze, SceneWalk and all extension versions that we introduce. On top of each prediction, we show the log-likelihood of how well the fixation is predicted. For the DeepGaze prediction, we additionally show the score difference between DeepGaze and SceneWalk that resulted in this fixation being picked as controversial. On top of the image we show the mean score difference that resulted in this image being picked.

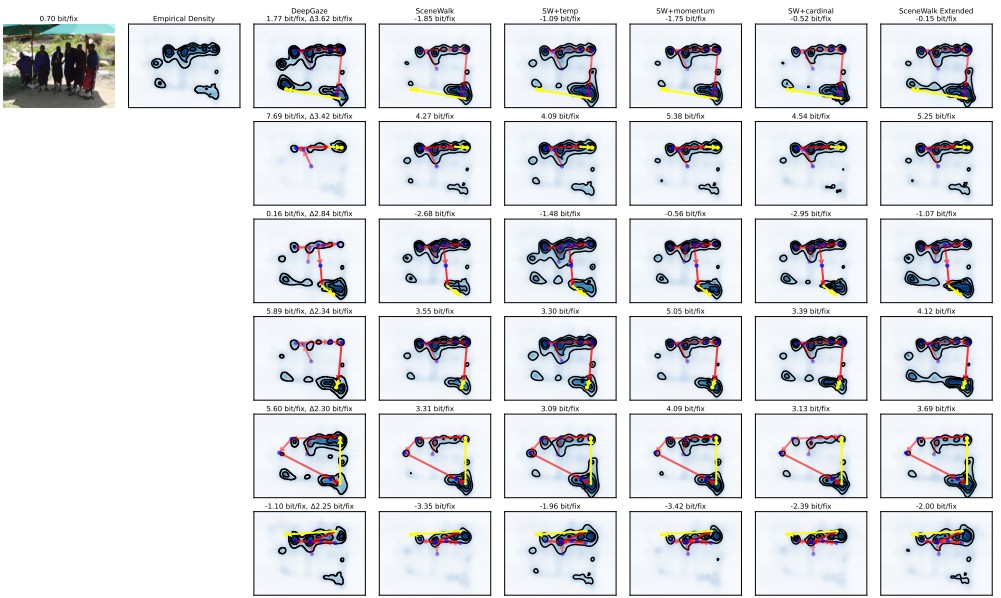

Figure 9: Controversial Fixations selected for maximum difference in log-likelihood between DeepGaze and SceneWalk, Part 2.

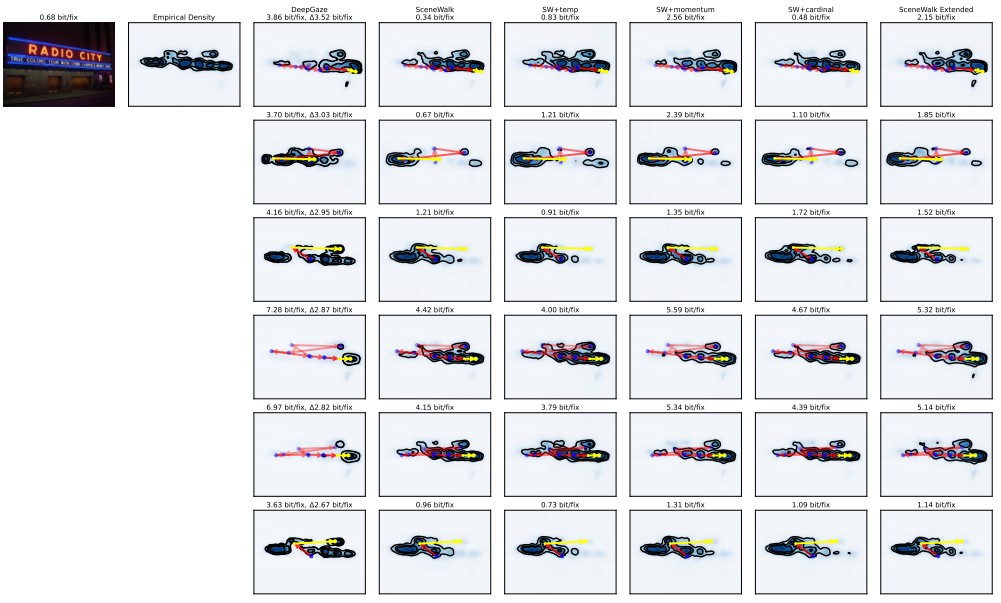

Figure 10: Controversial Fixations selected for maximum difference in log-likelihood between DeepGaze and SceneWalk, Part 3.

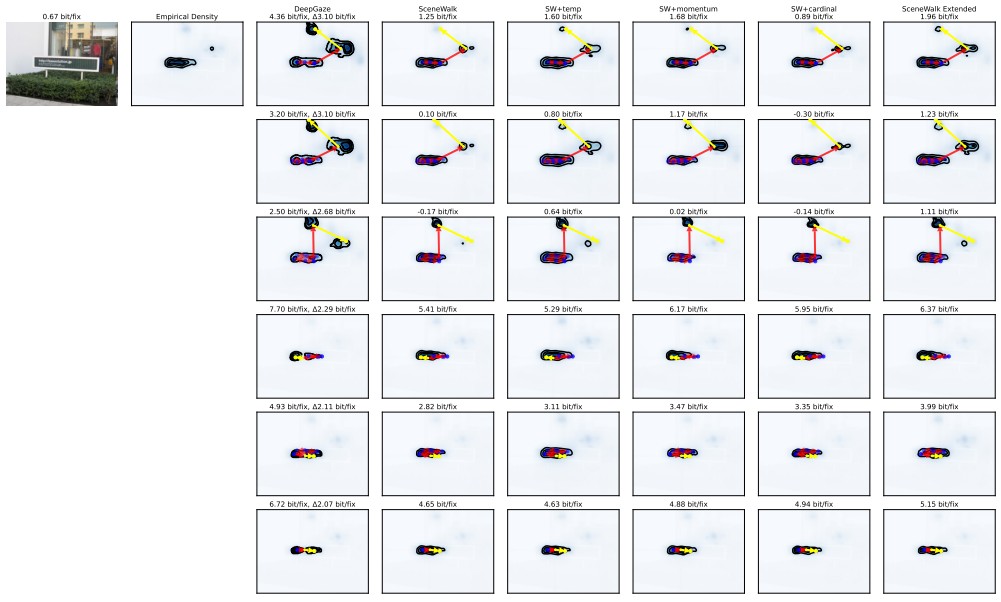

Figure 11: Controversial Fixations selected for maximum difference in log-likelihood between DeepGaze and SceneWalk, Part 4.

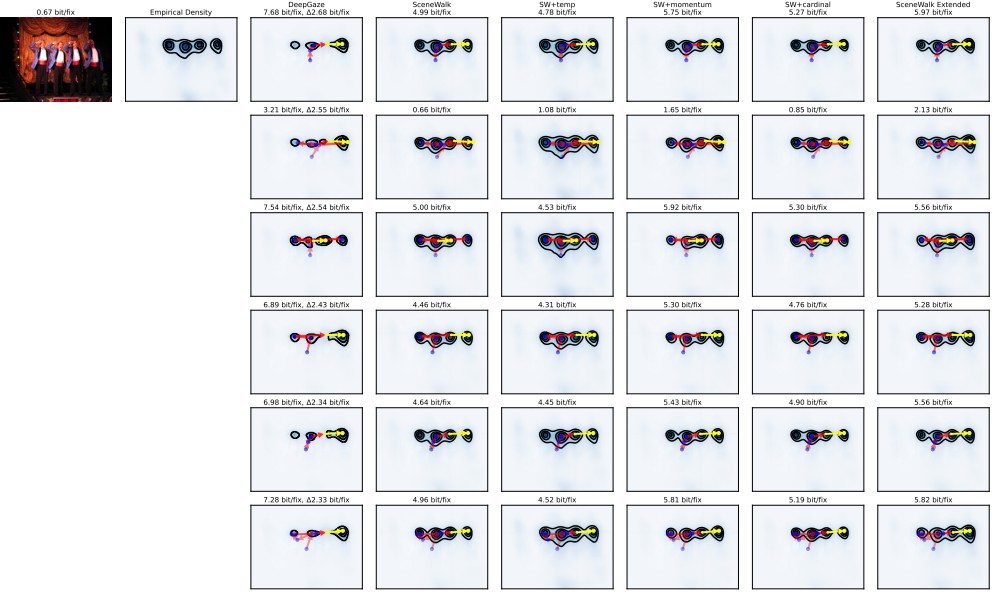

Figure 12: Controversial Fixations selected for maximum difference in log-likelihood between DeepGaze and SceneWalk, Part 5.

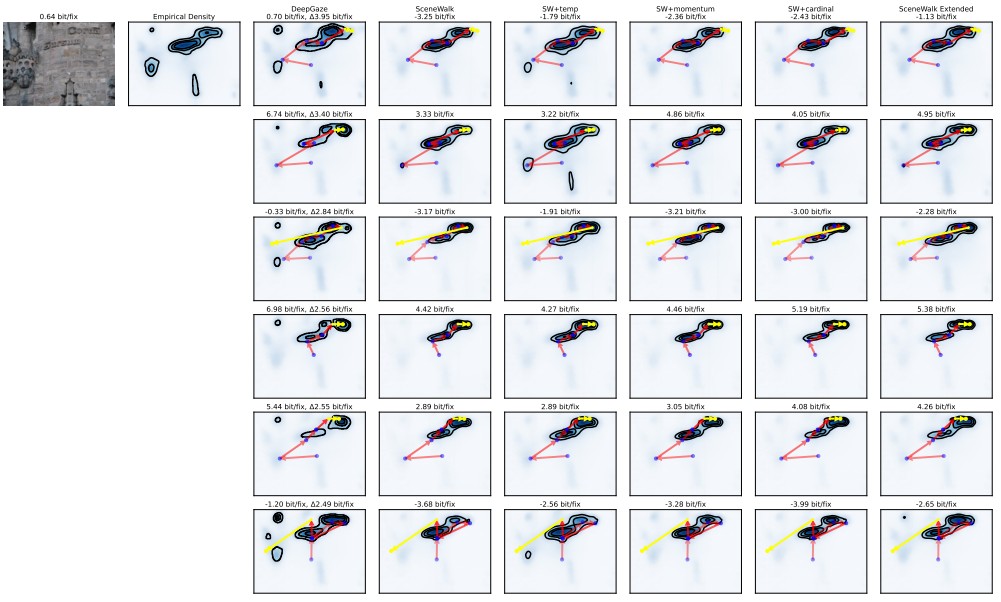

Figure 13: Controversial Fixations selected for maximum difference in log-likelihood between DeepGaze and SceneWalk, Part 6.

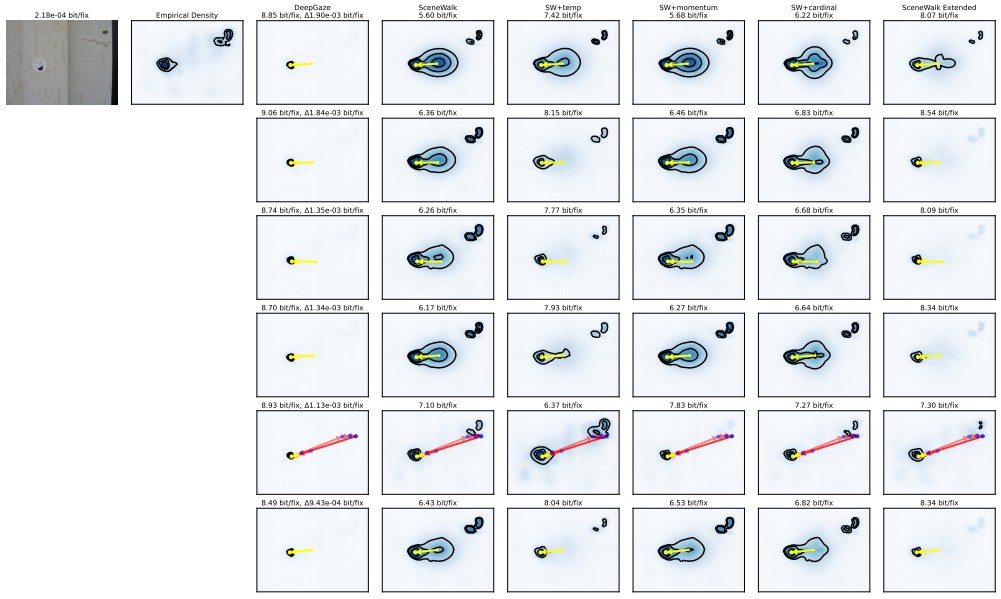

Figure 14: Controversial Fixations selected for maximum difference in log-likelihood between DeepGaze and SceneWalk weighted by the probability assigned by DeepGaze, Part 1.

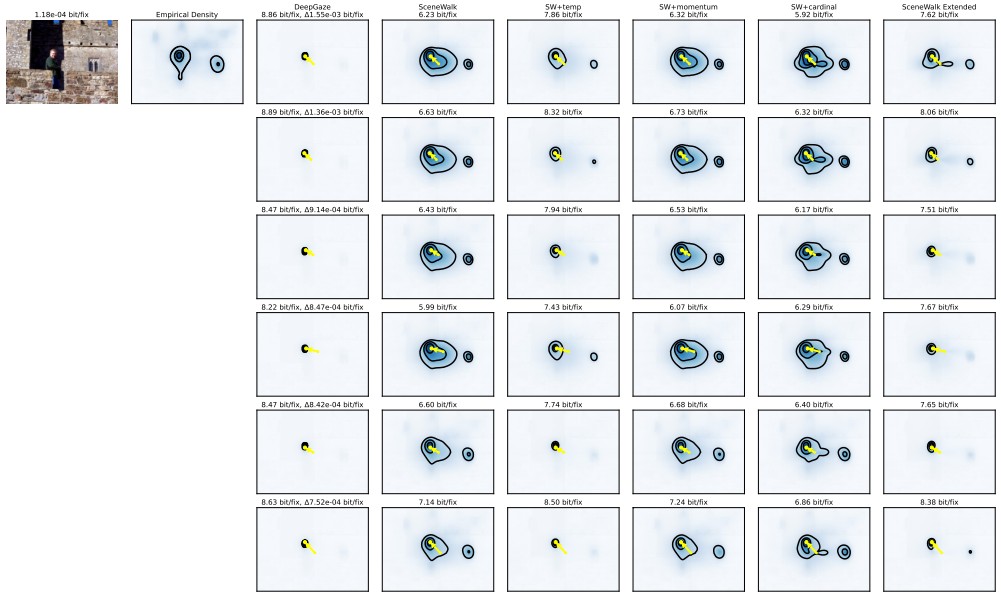

Figure 15: Controversial Fixations selected for maximum difference in log-likelihood between DeepGaze and SceneWalk weighted by the probability assigned by DeepGaze, Part 2.

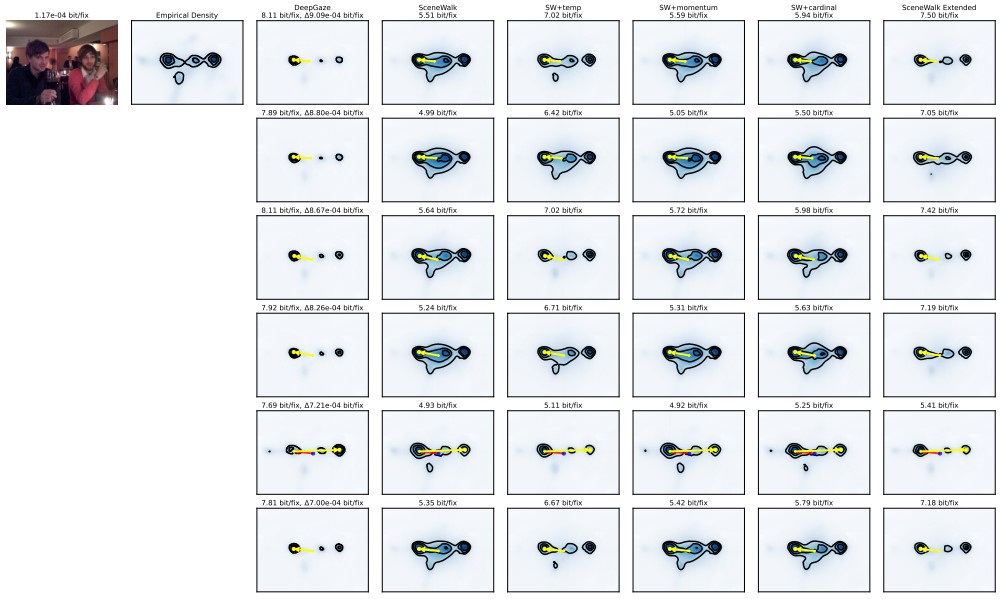

Figure 16: Controversial Fixations selected for maximum difference in log-likelihood between DeepGaze and SceneWalk weighted by the probability assigned by DeepGaze, Part 3.

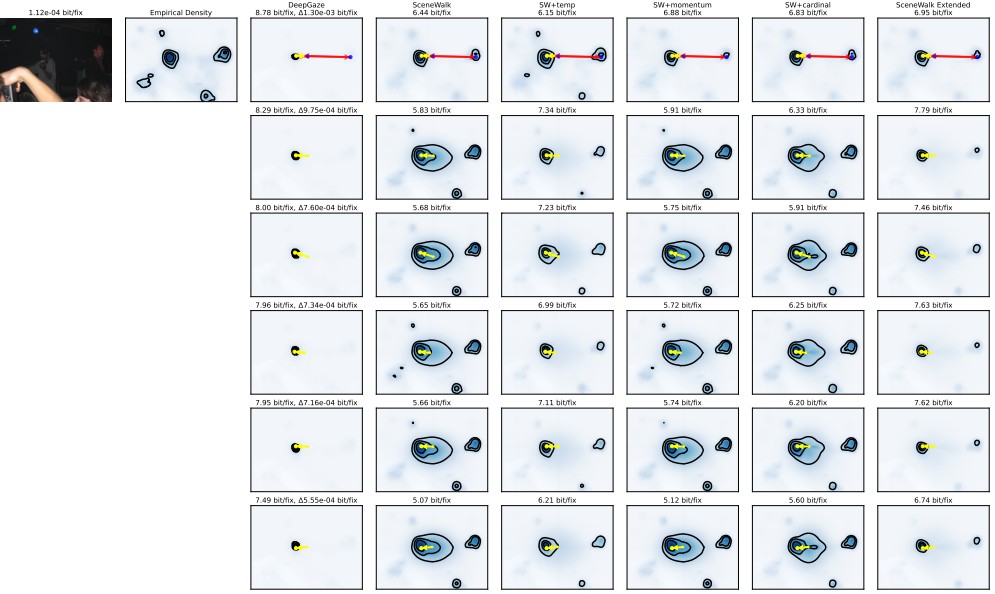

Figure 17: Controversial Fixations selected for maximum difference in log-likelihood between DeepGaze and SceneWalk weighted by the probability assigned by DeepGaze, Part 4.

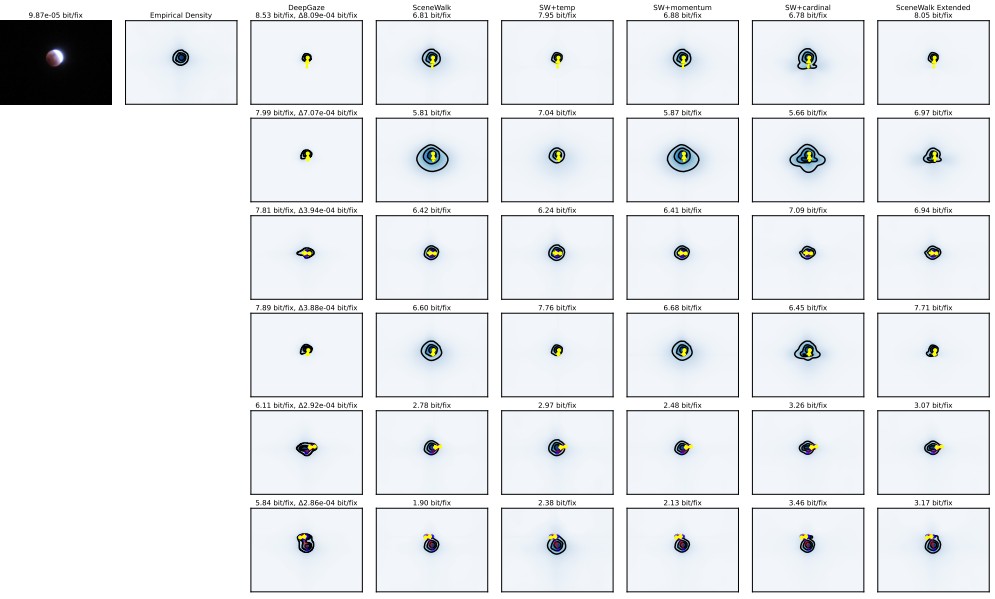

Figure 18: Controversial Fixations selected for maximum difference in log-likelihood between DeepGaze and SceneWalk weighted by the probability assigned by DeepGaze, Part 5.

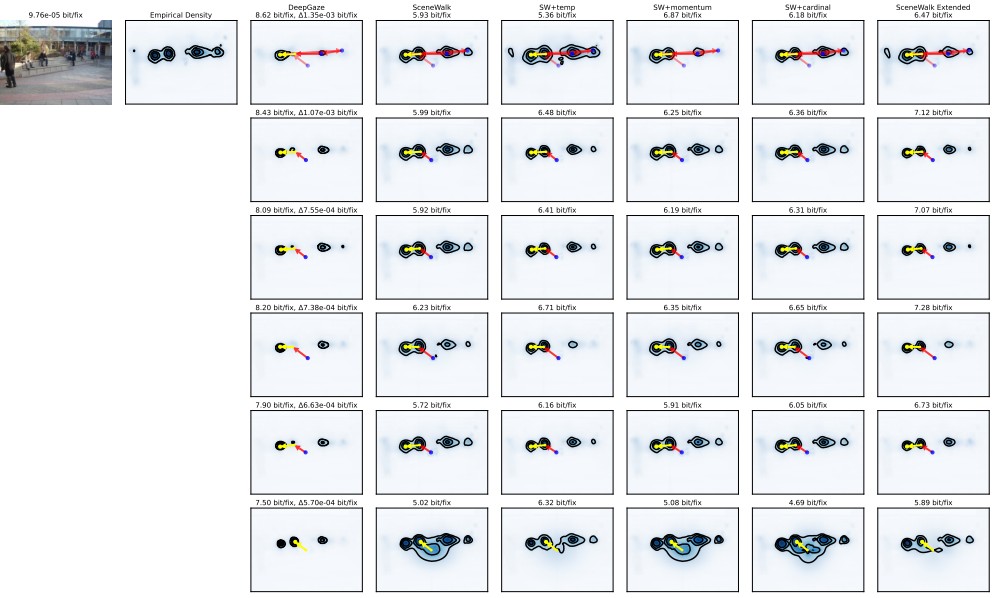

Figure 19: Controversial Fixations selected for maximum difference in log-likelihood between DeepGaze and SceneWalk weighted by the probability assigned by DeepGaze, Part 6.

