# OpenReview forum: "What Moves the Eyes: Doubling Mechanistic Model Performance Using Deep Networks to Discover and Test Cognitive Hypotheses"
_NeurIPS.cc/2025/Conference — NeurIPS 2025 poster_

### Official Review · Reviewer_SwGg · 2025-06-27

**Clarity:** 3
**Significance:** 3
**Originality:** 3
**Rating:** 4
**Confidence:** 4

**Summary:**

The authors propose to improve a mechanical model of eye movements on static images by using a data-driven model and to analyse the controversial predictions between both models in order to improve the mechanical model. The results show a significant improvement in the predicting eye movements.

**Questions:**

A major contribution of your model and establish a function of temperature and momentum that is introduced into the mechanistic model. Can you relate the parameters you changed to some biological data, whether mechanistic for example by considering the system of muscles that allow eyeball movement.

A major feature of the interaction between eye movements and scene understanding is that the image on the retina is foveated, thus allocating more ressources (photorecpetors) around the axis of sight. Could you relate your model to such models that would use foveated retinotopy?

**Ethical Concerns:**

["NO or VERY MINOR ethics concerns only"]

**Final Justification:**

While the responses to my concerns were clearly expressed, the proposed changes are too minor to reevaluate my score.

**Limitations:**

One limitation of the paper is its relatively incremental advancements, which, while valuable, do not significantly depart from existing methodologies. Interpretability would be a huge bonus.

**Paper Formatting Concerns:**

minor: You sometimes miss the space before a citation, eg "MIT300 benchmark dataset[38]". Some typos like "These effects reamain even "

**Quality:**

3

**Strengths And Weaknesses:**

The strength of the model presented here is that it presents a mechanical model, and is therefore much more realistic, biologically, which reduces the gap between models that perform well, but are unrealistic and are based on mechanical rules. The inferred statistics of the different characteristics of eye movements during the exploration of an image has particular significance for the scientific community studying  biological movements.

A weakness of the paper is that it is relatively incremental and shows improvement only mainly on the evaluation criteria that were used to learn the data-driven model.

---

> ### Author Rebuttal · Authors · 2025-07-31
>
> We thank you for your positive review and for your appreciation of our work’s goal.
> We are grateful for your feedback and your insightful questions, which will help us better connect our findings to the broader scientific context. Let us respond point by point below.
>
> ### [Question] On relating new mechanisms to biological data
>
> > A major contribution of your model and establish a function of temperature and momentum that is introduced into the mechanistic model. Can you relate the parameters you changed to some biological data, whether mechanistic for example by considering the system of muscles that allow eyeball movement.
>
> This is an excellent point. While our modeling is at a functional level, the mechanisms we've added are indeed motivated by and consistent with known biological and cognitive phenomena. However, exactly matching the parameter values is likely nontrivial at this point. For saccadic momentum, Wilming et al., (2013) show that muscle properties alone are not enough to explain the observed saccadic momentum and conclude that central oculomotor planning likely also plays a role. For the case of the temperature change, we expect a dependency on dataset statistics, task instructions and subject motivation.
>
> **Planned Revision:** We will expand our discussion of the different mechanisms to make these links more explicit. For saccadic momentum (lines 322, 341-342), we will reinforce its connection to established findings on oculomotor dynamics and facilitation-of-return. For the cardinal attention bias (lines 462-467), we will further highlight the literature suggesting such anisotropies may originate as early as the superior colliculus and reflect fundamental properties of attentional processing.
>
> ### [Question] On the relation to foveated retinotopy
>
> > A major feature of the interaction between eye movements and scene understanding is that the image on the retina is foveated, thus allocating more ressources (photorecpetors) around the axis of sight. Could you relate your model to such models that would use foveated retinotopy?
>
> This is a very insightful question. While SceneWalk does not simulate a foveated retina at the pixel level, its core principles are compatible with the functional consequences of foveation. The model does not operate directly on the image itself, but rather on a static saliency map derived from the image. This map is initially hidden and we uncover it locally by looking somewhere, which opens a Gaussian window on the saliency map: This resembles how visual acuity is concentrated at the center of gaze. In this sense, SceneWalk shares important functional similarities with models that explicitly incorporate foveated retinotopy (such as through image-space transformations) though it operates at a higher level of abstraction, dynamically allocating resources over a saliency map rather than a pixel-level retinal representation.
>
> Going one step earlier and exploring in more detail how scene content is integrated into a latent state that is then used to select fixations would be a very exciting next step. A conceptually simple approach would be one one from MASC (Adeli et al., 2017), where peripherally blurred images are passed to a saliency backbone to update an internal saliency map. More appropriate but also more involved would be a backbone that directly models foveation, e.g., FLIP (Traub et al., 2025), and could include object attention, semantic attention, scene relationships and other effects.
>
> **Planned Revision:** We will add a sentence to the discussion acknowledging this strong conceptual link. We will state that our model captures the functional dynamics of foveated vision and suggest that incorporating an explicit foveated input representation is a promising avenue for future work, both for SceneWalk and for DeepGaze III.
>
> ### [Weakness] On scope of the contributions
>
> > A weakness of the paper is that it is relatively incremental and shows improvement only mainly on the evaluation criteria that were used to learn the data-driven model.
>
> While we agree that our approach builds on existing components, we believe its value lies in demonstrating how these components can be systematically integrated to advance both mechanistic modeling and interpretability in deep learning. Our framework is inspired by the idea of scientific regret minimization (which we cite and build upon), but we go beyond prior work by applying it in a concrete setting where we substantially improve a mechanistic model and simultaneously gain insight into the inductive biases of a deep network.
>
>
> We view this as a contribution of both methodological and conceptual significance: it shows how existing ideas can be operationalized to generate interpretable, testable refinements to biologically inspired models. While indeed the improvements made to SceneWalk may be incremental in a narrow technical sense, we believe such work is crucial for bridging the gap between high-performing performance-driven models and mechanistic understanding, a central challenge in both neuroscience and AI.
>
> Finally, we want to emphasize that the performance gain we achieve is not small, but closes a substantial part of the gap between purely mechanistic performance and DNN performance.
>
> ---
>
> Thank you again for the constructive review. We hope our responses have clarified how this work, while building on prior ideas, provides a concrete path toward improving mechanistic models by leveraging insights from deep networks. Beyond predictive performance, we believe the framework contributes to a growing effort to build models that are both interpretable and grounded in biological principles—an important step for bridging data-driven and theory-driven approaches to understanding behavior. We hope our response has helped support your positive assessment or even strengthened your confidence in the paper’s value.

---

> > ### Author Response · Authors · 2025-08-06
> >
> > We want to add one more detail regarding one of the weaknesses the reviewer mentioned:
> > > shows improvement only mainly on the evaluation criteria that were used to learn the data-driven model.
> >
> >
> > we now also evaluated AUC, which is one of the most commonly used metrics in eye movement prediction. We consider it nicely orthogonal to our main metric "log-likelihood" because while LL is sensitive to every detail of the predicted distribution, AUC is very robust by only caring about the rank order of the distribution. This makes it a very qualitative metric that is good at assessing whether the most pronounced features of a distribution have been captured. Here are the results for MIT1003:
> >
> > | model            |      AUC | % explained |
> > |:-----------------|---------:|-----------:|
> > | Spatial baseline | 0.922293 | 0.000          |
> > | SceneWalk        | 0.930437 | 0.501   |
> > | SceneWalk-X      | 0.934264 | 0.736   |
> > | DeepGaze         | 0.93855  | 1.000        |
> >
> > We see that our main result also holds in AUC: our additional mechanisms close about half of the gap in performance between SceneWalk and DeepGaze. The base scenewalk model performs a bit better in AUC than in LL at about 50% of the performance gap between the spatial baseline and DeepGaze, compared to 30% in log-likelihood. We will add this result, as well as AUC scores for DAEMONS and the Potsdam dataset to the paper.

---

### Official Review · Reviewer_i2u5 · 2025-06-30

**Clarity:** 2
**Significance:** 2
**Originality:** 2
**Rating:** 3
**Confidence:** 4

**Summary:**

This paper uses a deep neural network (DNN) model of human gaze behavior called DeepGaze III (DG III) to improve a mechanistic model called SceneWalk. In particular, the authors applied both models to the MIT1003 dataset of images and determined specific cases where the scanpath predictions of DG III and SceneWalk differed. Then, the authors used those "controversial fixations" to come up with extensions and modifications to SceneWalk that would make the model perform more similarly to DG III. (In other words, they identified common patterns that explained the different fixation predictions made by the DG III and SceneWalk so that they could generate new mechanisms to add to SceneWalk that fit those patterns.) In particular, they added the following features to SceneWalk: time dependent temperature scaling (favoring high-saliency targets early in scanpaths), saccadic momentum (from an oculumotor bias map), and a horizontal and left-ward bias. These modifications brought SceneWalk's performance from 0.15 bit/fix to about 0.3 bit/fix, compared to DG III's 0.45 bit/fix.

**Questions:**

To make the rebuttal process more constructive, I’ve aligned my questions below with the corresponding numbered weaknesses in my review.

1a. Would the authors consider revising their conceptual framing to reflect the growing use of task-performing DNNs as mechanistic models (e.g., surprise-driven DNNs, goal-directed architectures, or task-constrained pretraining)? This would allow their discussion to better integrate and contextualize recent trends in computational cognitive modeling.

1b. Can the authors justify their choice to retrofit SceneWalk using DG III outputs, rather than analyzing DG III itself to extract mechanistic principles (which they show is possible)? What is gained by working through a weaker model rather than analyzing the stronger one directly?
The paper generates mechanistic insights using discrepancies between SceneWalk and DG III, but the original DG III paper also reports its own mechanistic insights. Could the authors clarify why their methodology brings something unique that mechanistic analyses of DG III don't?

1c. Could the insights the authors take from this paper ultimately be applied to mechanistic DNN models? Could we aim for an explanatory model that exceeds DG III?

2. Can the authors provide any estimate of human inter-observer consistency as a comparison point? Or, alternatively, justify in more detail why DG III should be treated as a stand-in for human scanpaths?

3. Could the authors clarify the data partitioning at each stage of their pipeline? Specifically:
- What data is used for initial SceneWalk vs. DG III comparisons?
- How is the final performance comparison between SceneWalk vs. DG III done: is it on held-out data?
- Is there any data from human performance to demonstrate that SceneWalk is now more like human scanpaths and not just more like DG III?

A note on limitations of possible revisions:
For the sake of transparency, I want to be clear that even if the authors address all the questions above, my core concern may still remain. The central methodological premise — retrofitting a simpler model to approximate a stronger one that already exists and performs better — seems conceptually inefficient and hard to justify. Unless the authors can articulate why this retrofit strategy is necessary or uniquely revealing, my evaluation may not shift substantially, even if surface-level issues are resolved.

**Ethical Concerns:**

["NO or VERY MINOR ethics concerns only"]

**Final Justification:**

The rebuttal was exceptionally clear, resolved my methodological concern about circularity, and clarified the authors’ position on the continuum between interpretable and performance-driven models. My main reservations remain philosophical rather than technical — I continue to view directly analyzing or refining high-performing DNNs as a more promising path than retrofitting simpler mechanistic models. Having said that, I do see merit in different modeling philosophies coexisting in the field. Overall, my score reflects my evaluation of the paper's alignment with my acceptance threshold, not the quality of the exchange, which was excellent.

**Limitations:**

Potential limitations are already addressed above. I see little (if any) potential negative societal impact of their work.

**Paper Formatting Concerns:**

Minor typo issues:
- Missing carriage return line 27.
- Missing “we” line 45
- Extra space line 185
- Spelling strengths line 305
- Spelling quantitatively line 282

**Quality:**

2

**Strengths And Weaknesses:**

Strengths:

This paper tackles a central question in cognitive science and computational modeling: what drives visual attention? This question is critical for understanding vision writ-large because what we attend to becomes the training curriculum for visual learning in general. The authors engaged with the literature on scanpath modeling and positioned their work in relation to both cognitive models and deep learning approaches. This is a promising direction, but I'm not yet convinced by the current version.

Weaknesses:

1) The paper sets up a false dichotomy between mechanistic models and DNNs. Deep networks are not inherently non-mechanistic; their internals can embody explicit cognitive hypotheses (e.g., energy-based surprise models for gaze—Aakur & Bagavathi 2020). By equating “mechanistic” with “non-DNN" the authors ignore the possibility of leveraging deep models that they already recognize as more powerful models. Ironically, the paper is able to generate 3 mechanistic insights from the performance of DG III, and the original DG III paper (Kummerer et al., 2022) includes multiple mechanistic insights like interplay of scene content and scanpath history.

More broadly, DNNs are becoming increasingly popular as formal, task-performing models of cognition. If a DNN can perform the same task as the brain, then this serves as evidence that the brain may solve the task in a similar way. The brain is a complex system of billions of neurons, and its operations may very well be better characterized by an optimization process (ala deep learning) than by simple cognitive rules. Moreover, while the authors are correct that most deep learning approaches to scanpaths (e.g., DG III) learn to predict human fixation directly, DNNs do not have to be trained this way by necessity. It's possible to train DNNs with the same data as humans (i.e., without supervised human behavior like image labels and scanpaths) to test if they can learn the same skills from the same types of data.

The result of this false dichotomy is that the authors acknowledge how much more powerful DNN models are, but rather than embracing DNNs and using DNNs directly to learn about scanpath mechanisms, they use DG III to bootstrap SceneWalk. This is analogous to having access to a Ferrari (DNN) and choosing to retrofit a horse-and-buggy cart (cognitive model) to follow its tracks. Rather than embracing the capabilities of the more powerful vehicle, they attempt to enhance the older model—ultimately limiting the practical gains that could be realized by simply using the better tool.

2) It's not clear why the paper does not include a human upper bound on performance. The only explanation offered is that "estimates of inter-observer consistency are hard to derive" "in high-dimensional distributions." However, I don't understand why that difficulty would not apply equally to comparing any two high-dimensional distributions (including model-vs-model and human-vs-model). Instead the paper uses DG III as gold standard / ground truth, but without evidence that DG III performance should be trusted as ceiling.

3) Relatedly, by using DG III both as the gold standard and to find controversial fixations, the method becomes somewhat circular. The overall approach is: (1) contrast SceneWalk with DG III, (2) hand-engineer changes to reduce the gap, (3) measure whether SceneWalk now more closely resembles DG III. Because performance is never validated against human behavior, the exercise risks becoming self-referential. Instead, it would help for the authors to show that changes which shrink the SceneWalk-DG III gap also shrink the gap to humans on a held-out images.

Another change that is also needed to deal with possible circularity issues is to specify (in SI) whether the data used to diagnose SceneWalk-DG III discrepancies is disjoint from the data used to evaluate the final, tuned model. There is some limited discussion about crossvalidation folds in SI, but it's not detailed enough for me to tease apart the stages of training, testing, initial SceneWalk-DG III comparison, and final SceneWalk-DG III evaluation.

---

> ### Author Rebuttal · Authors · 2025-07-31
>
> Thank you for your deeply engaged and challenging review. We appreciate the time you took to analyze our work from a conceptual standpoint. Your feedback has pushed us to realize that our framing of the relationship between mechanistic models and DNNs was not as nuanced as it should have been, and we are grateful for the opportunity to clarify our methodology and contributions. We will start with your main concern about the usefulness of “retrofitting” simple models, and then continue with the other questions and issues.
>
> ### [Q1b] Why simple models?
>
> The question touches on fundamental differences in modeling philosophy: namely, what counts as scientific understanding. We respectfully argue that even in the eara of of powerful function approximators, simple mechanistic models are still essential in cognitive science, and dismissing their widespread and long-standing role across the sciences altogether reflects a perspective that is not representative of the field as a whole.
>
> At the core of our perspective is the idea that scientific understanding essentially is a form of compression: expressing complex phenomena, like gaze behavior, in terms of a smaller set of testable and generalizable principles, typically using language (natural or optimally mathematical). From this view, shared by many in philosophy of science (e.g., Wilkenfeld, 2019), a DNN with millions of entangled parameters constitutes a much less efficient description than a mechanistic model with a handful of interpretable components. All else being equal, the latter better supports explanation, communication, and theory-building, in our case being clearly positioned in the algorithmic level out of Marr's (1982) famous three levels (computational, algorithmic, and implementational). Of course, simple explanations are not helpful when they make terrible predictions: Ultimately, there is a pareto front along the tradeoff between simplicity and predictive power. We consider it worthwhile to also (not only!) strive for the most simplistic but still predictive account of behaviour, which forces us to explain how different interpretable mechanisms interact. This is in line with long standing and still active traditions in the field (Craver et al 2024’s “Mechanisms in Science” in the Encyclopedia of Philosophy; Bechtel’s mechanistic philosophy, 2011).
>
> We understand the concern about “retrofitting” a simpler model to match a more powerful one. However, our goal is not to retrofit in a superficial sense, but to use the predictive success of DG as a scaffold for structured hypothesis generation. Rather than digging into the internals of the DNN – which, when dealing with cognitive phenomena, can be prone to pitfalls (e.g. Geirhos et al., 2020) – we ask: What kinds of behavioral errors remain in the mechanistic model, and do they point to missing cognitive components? The value of this approach is backed by recent publications in top‐tier venues like Nature, which, like the Centaur foundation model—explicitly position computational predictions as stepping stones toward theory: they argue that predictive models offer ‘tremendous potential for guiding the development of cognitive theories’. This synergy is also strongly endorsed in the philosophical literature (e.g. Harbecke 2021) and empirical practice in cognitive science.
>
> As a final point, we’d like to playfully return to the car analogy. SceneWalk-X achieves over 70% of DeepGaze III’s performance using fewer than 1% of its parameters (31 vs. thousands) and compute. In that light, it is at least intriguing, we argue, that a horse-and-buggy can reach 70% of the Ferrari’s top speed while using <1% of the fuel. And while you might pick a Ferrari to go fast, if your goal is to teach someone how a car works you might prefer the old-timer with the hood open and the individual components clearly visible.
>
> ### [Q1a] On the continuum between mechanistic models and DNNs
>
> We agree that our framing should more clearly reflect the continuum of approaches between classic cognitive models and modern deep networks. While DNNs are indeed computational mechanisms, and can instantiate cognitive hypotheses or be trained with elegant constraints, their complexity typically prevents full interpretability. Our use of the term “mechanistic model” (grounded in the modelling literature in neuroscience and cognitive science) refers specifically to models built with transparent, modular components that directly reflect hypotheses. However, we acknowledge that between the extremes, a continuum of hybrids and more interpretable DNNs has evolved with different tradeoffs between performance and simplicity. We will revise our terminology to reflect this continuum more explicitly and justify our position therein.
>
> Further, we fully agree with the reviewer that DNNs can be integrated into scientific explanations when their structure is designed with clear cognitive hypotheses in mind. The scene-history interaction component of DeepGaze III is indeed such an example and shows that DNNs can yield individual mechanistic insights.
>
> Yet, we consider these orthogonal but related approaches at different ends of the above-mentioned pareto front: In DeepGaze III, a high-performing model is made more interpretable, while we aim at making a fully interpretable model more high-performing.
> The former establishes that a mechanism exists by asserting that we can't predict behaviour better by loosening constraints. The latter establishes how much of the overall behaviour we can explain via only fully interpretable mechanisms.
>
> **Planned revision (Q1a and Q1b):** We will revise and extend our introduction and discussion to clearly reflect the continuum between interpretability-first and performance-first models. Specifically, we will (1) clarify our use of “mechanistic,” (2) acknowledge the increasing role of hybrid and task-constrained DNNs in cognitive modeling, and (3) better articulate why interpretable models remain crucial for scientific understanding, even when performance is not maximized.
>
> ### [Weakness] On concerns of circularity
>
> We agree that care must be taken to avoid circular reasoning when using one model to guide improvements in another. However, we believe the core of the issue raised might stem from a misinterpretation of how our evaluation is performed.
>
> While we do use DeepGaze III to identify controversial fixations, we do not measure the success of our method by checking whether SceneWalk becomes more similar to DG. Instead, we evaluate all model variants using log-likelihood scores *against human fixations*, i.e., real behavioral data. Our metric directly reflects how well the model predicts actual human eye movements, not whether it agrees with DG III. In this context, DG, in the form of its performance, just provides an estimate for the log-likelihood on the true data that *could* be achieved.
>
> Furthermore, we believe the inclusion of a second, completely held-out dataset (DAEMONS), where SceneWalk-X also improves, provides strong evidence that our methodology leads to genuine improvements in human behavioral prediction.
>
> ### [Q1c]
> > Could the insights the authors take from this paper ultimately be applied to mechanistic DNN models? Could we aim for an explanatory model that exceeds DG III?
>
> We agree this is a compelling direction. As long as the DNN model is sufficiently powerful and there is enough data to train it, it should always perform at least as well as a more constrained model. Given that data is not infinite, though, it might be possible to find mechanisms that given the possible DNN size are hard to approximate and where a direct implementation could perform better, especially on OOD data.
>
> ### [Weakness + Question 2] On upper bound explainability estimates
>
> We appreciate this important point, which was also raised by reviewer 3tRx. Please check our response there for a more detailed explanation and our revision plan for addressing this clarity issue.
>
> Briefly, each fixation must be predicted from a unique and growing history of previous fixations, so the relevant distribution depends on up to ~20 input variables per step. With only about 100 fixations per image, this leads to extremely sparse coverage, making empirical or nonparametric baselines infeasible.
>
> ### [Question 3] Data partitioning and evaluation
>
> For model training and evaluation, we used 10-fold cross-validation on MIT1003 over images, for each fold using 8 parts for training, one part for analysis and one part for final performance evaluation, ensuring that both DG and SW were always evaluated on held-out data with respect to fitting.
>
> * The initial DG vs SW comparisons and the identification of controversial fixations were done across the full MIT1003 dataset, looking at conditional densities on *validation splits* for a particular image.
> * The final performance on MIT1003 (Fig. 1b and 6) is reported aggregated over *test splits*. For DAEMONS, which was fully held out during development, it is on the dataset’s provided public validation split.
> * We evaluate models using log-likelihood scores against human fixations, i.e., real behavioral data. Our metric directly reflects how well the model predicts actual human eye movements, not whether it agrees with DG III.
>
> ---
>
> We thank you once more for your review, which prompted us to sharpen our conceptual distinctions and more explicitly justify key design choices. We also appreciate the opportunity to clarify that our evaluation is always against human data, and that we avoid the risk of circularity through cross-validation and out-of-distribution testing.
>
> In light of these clarifications, we hope you might view the work not only as an interesting idea, but also as a carefully executed and generalizable contribution. In spite of your confidence in the original judgment, we respectfully invite you to reconsider whether the manuscript might warrant a more favorable assessment.

---

> > ### Comment · Reviewer_i2u5 · 2025-08-06
> >
> > First, I want to sincerely thank the authors for engaging in such a thoughtful and stimulating theoretical discussion. I found the rebuttal to be clear, well-argued, and genuinely enjoyable to read. I especially appreciate the authors’ willingness to engage directly with the conceptual framing of their work. I agree that the core disagreement reflects fundamental differences in modeling philosophy, and I appreciate their recognition of that.
> >
> > To be transparent, I don’t share the authors’ modeling philosophy. I believe that cognitive science often overvalues simplicity in a way that limits theoretical progress. There’s a persistent tendency to favor small, hand-tuned models as “more explanatory” simply because they are easier to interpret. Meanwhile, DNNs are too often dismissed as black boxes or seen as insufficiently mechanistic, despite their increasing ability to match or exceed human-level performance and instantiate sophisticated cognitive processes.
> >
> > In that light, I still find it hard to *fully* endorse the approach of retrofitting a cognitively inspired model to match a stronger DNN, rather than directly analyzing or refining the DNN itself. I really appreciate their clever car analogy (and thank you for running with my cheeky metaphor!), but I think it ultimately feeds into my point: if the goal is to teach a novice how a car works, then yes, the old-timer with the hood open is better. But as scientists, our goal should go beyond building explanations that are merely novice-accessible. The brain is a dynamic, high-dimensional system, and understanding it likely requires tools that exceed what we can verbalize or intuit through simplified rules. DNNs may be complex, but they offer the possibility of capturing emergent behaviors and properties that are fundamentally inaccessible to small, modular models.
> >
> > Having said that, I strongly agree that these are the kinds of debates our field should be having. I appreciate the authors’ openness to reframing their terminology to better reflect the continuum of modeling approaches, from fully interpretable cognitive models to high-performing deep networks. I also value their clear articulation of the pareto front between interpretability and performance, which is a helpful framing for thinking about how different models contribute to cognitive science.
> >
> > On a more concrete point: I want to thank the authors for clarifying the evaluation procedure, especially in regard to circularity. I now fully understand that all models were evaluated against held-out human fixations, not DeepGaze III predictions. That completely resolves my concern — and I apologize for my earlier misunderstanding. I appreciate the careful explanation.
> >
> > In light of this helpful discussion, I do plan to increase my score accordingly. Thank you again for the thought-provoking exchange!

---

> > > ### Author Response · Authors · 2025-08-08
> > >
> > > We thank the reviewer for their thoughtful follow‐up and for engaging in such a stimulating exchange. We greatly enjoyed the discussion, which helped us to sharpen how we articulate the relationship between our work and other modeling approaches, as well as our broader view on scientific understanding. We think regarding our submission nothing more needs to be said, but there is one thought that we want to add to our discussion:
> > >
> > > > The brain is a dynamic, high-dimensional system, and understanding it likely requires tools that exceed what we can verbalize or intuit through simplified rules. DNNs may be complex, but they offer the possibility of capturing emergent behaviors and properties that are fundamentally inaccessible to small, modular models.
> > >
> > > We fully agree with this assessment and we also recognize the tendency in some areas to overemphasize overly simplistic models. In our view, simple models are helpful when they can account for a substantial share of the variance in behavior - which is not always the case. And even if they are, the residual variance is likely due to complex dynamics that cannot be easily expressed in a compact mechanistic form. DNNs provide a powerful way to address this problem and we are happily using them in these cases.
> > >
> > > When starting to work on the current project, we actually expected something like this to happen: we considered it quite likely that the controversial fixations will be due to a myriad of different, subtle effects and that we won't be able to extract clear patterns from them. When we analyzed the controversial fixations, we were somewhat surprised by how clearly we could see missing patterns. As a result of our discussion with the reviewer, we think it might be worth emphasizing more explicitly in the paper the paper that this outcome was not obvious from the onset.

---

### Official Review · Reviewer_DPB9 · 2025-07-01

**Clarity:** 4
**Significance:** 3
**Originality:** 2
**Rating:** 5
**Confidence:** 5

**Summary:**

In this manuscript the authors improve the mechanistic SceneWalk model which predicts scan paths of humans free viewing natural scenes. They base these improvements on comparisons to DeepGaze III, a deep learning model for the same task, which yields “controversial fixations”, I.e. fixations where the deep learning model still performs much better than the mechanistic one. Concretely they add time dependent temperature scaling to make the predictions higher entropy later in the trial, saccadic momentum to implement return saccades and the tendency to continue moving in the same direction, and an attention bias towards the cardinal directions and horizontal directions in particular. Overall these additions improve the prediction quality of the SceneWalk model by a substantial 0.2 bits/fixation on the DAEMONS and MIT1003 data sets.

**Questions:**

The weighted log-likelihood difference is an unusual measure for the difference between predictions. Why do the authors prefer it over a more classical measure like the KL-divergence?

Wouldn’t it be interesting to evaluate the model additions on the data used originally to develop the SceneWalk model? They are available (https://osf.io/n3byq/)

**Ethical Concerns:**

["NO or VERY MINOR ethics concerns only"]

**Final Justification:**

Unsurprisingly, the authors response did not change my opinion about this paper much.

Reading through the other reviews and responses also did not change my rating. I would now give the authors a little more novelty credit, because this way of using DNNs to improve models clearly had not reached all the reviewers of this paper yet. At the same time, some of the misunderstandings hint at the paper seeming more clear to me than it actually was due to my prior knowledge. Keeping things as they are seems fitting to me.

**Limitations:**

yes

**Paper Formatting Concerns:**

none, but some of the figures are too small for reading in print

**Quality:**

3

**Strengths And Weaknesses:**

Overall, this is solid work. The authors clearly improve the model by comparing to deep learning predictions and also implement the SceneWalk model in JAX which doubtlessly will improve speed, too. Also, the additions to the model are all sensible and correspond to regularly observed data patterns in scan paths. However, this does imply that the comparison to DeepGaze might not have been strictly necessary and comparing to deep learning models to improve mechanistic ones is also not a new idea. To their Credit the authors acknowledge and cite the earlier occurrences of these ideas. Thus, I overall like this work, but I do not believe it is a big breakthrough.

In this manuscript the authors add a number of interpretable new mechanisms to the state of the art mechanistic model of eye movements. While I think specialist experimentalists in this particular subfield could know about the existence of the modelled phenomena, there are no mechanistic models that contain the added effects. Thus, this work really moves the state of the art forward for this type of mechanistic eye movement model. Also the path chosen to implement this improvement is a fairly recent addition in cognitive modelling toolkit, which I have not seen applied to eye movement models anywhere before and the authors promise to make their expanded model and a faster version of the original one public. So this paper is---in my view---a substantial and noteworthy step forward for models of eye movement trajectories.

Also, I like that this article does actually appear solid. For each of the new mechanisms there is evidence in the data and the authors know of corresponding literature and cite it. This makes this an exceptionally clear and solid paper. This solidness is also the reason for my fairly short review. I read through the descriptions of all the different mechanisms and found them to be generally sensible. If they were exceptional ideas or questionable, I would have had much more to write about, but they are all fairly straightforward implementations of the observed phenomena.

I value solid research highly and believe this paper does well on this front, which is why I vote for acceptance even though I have to acknowledge that the parts that the authors put together here have been presented elsewhere before, such that an argument for originality or novelty is hard to make.

---

> ### Author Rebuttal · Authors · 2025-07-30
>
> Thank you for your thorough and positive review. We are very grateful for your support of our work and for your encouraging assessment of its solidity and contribution to the field. We also appreciate the questions you raised, which will help us further refine the paper.
>
> ### [Question] On the use of Weighted Log-Likelihood Difference (WLLD)
>
> > The weighted log-likelihood difference is an unusual measure for the difference between predictions. Why do the authors prefer it over a more classical measure like the KL-divergence?
>
> You raise an important point. Indeed, KL divergence is a widely used measure for comparing distributions, which we have considered as it could in principle be applied here. However, we chose the Weighted Log-Likelihood Difference (WLLD) for two main conceptual and practical reasons.
>
> Conceptually, WLLD better reflects our analysis goals: It emphasizes how well a model predicted the actual observed fixation, rather than comparing the full predicted distributions. This aligns with our focus on understanding behaviorally relevant errors. By weighting the log-likelihood difference by DeepGaze’s predicted probability, we highlight fixations that DeepGaze deems highly likely but where SceneWalk fails. It is these cases that are especially informative for model improvement and interpretability purposes. This is more flexible than what standard KL divergence prioritizes.
>
> Practically, WLLD is far more efficient to compute and store. Because we score models using log-likelihoods at the ground-truth fixation locations, we only need to store one scalar per fixation per model. In contrast, computing KL divergence between the full distributions would require saving the complete per-fixation prediction maps, which can be prohibitive in terms of memory: over 300 GB for MIT1003 alone if saved without compression (100,000 fixations * 1024px * 768 px * 4 bytes). Since our method involves computing differences across many models and fixations, this would have been less manageable at scale.
>
> On a conceptual level, WLLD and KL-Divergence are actually closely related: KL-Divergence in our case would be a pixelwise sum over weighted differences in log probability. Our WLLD essentially only takes the value from the pixel that actually has been fixated next.
>
> **Planned revision:** We will clarify this rationale in the final version, and we agree that exploring alternative divergence measures could be an interesting future extension.
>
> ### [Question] On evaluating on the Potsdam dataset
>
> > Wouldn’t it be interesting to evaluate the model additions on the data used originally to develop the SceneWalk model?
>
> We thank the reviewer for this thoughtful suggestion. We agree that evaluating on the original dataset used to develop the SceneWalk model provides a valuable reference and an additional test of generalization. Following this suggestion, we ran the updated SceneWalk-X model and individual new mechanisms on the Potsdam corpus.
>
> | Model                    |   Information Gain |
> |:-------------------------|-------------------:|
> | Spatial baseline         |            1.80648 |
> | SceneWalk (classic)      |            2.23223 |
> | SceneWalk (perisaccadic) |            2.32155 |
> | SceneWalk-X              |            2.42571 |
> | DeepGaze                 |            2.54    |
>
> The results confirm the same qualitative pattern shown when evaluating on DAEMONS in Figure 6: temperature scaling yields modest improvements when averaged across the dataset due to the longer viewing times in this corpus compared to MIT1003, but the overall SceneWalk-X model improves predictive performance over the original perisaccadic SceneWalk by 0.1 bits/fixation (a 20% improvement with the static baseline as a reference), and closes approximately 47% of the remaining performance gap to DeepGaze III (reaching 82% “performance explained” in absolute terms as used in Figure 6). These findings further support the generality and robustness of the model extensions we propose.
>
> **Planned revision:** We will include these additional evaluation results in the appendix of the revised version of the paper.
>
> ### [Formatting concern] On the readability of figures
>
> Thank you for pointing this out. We will carefully review all figures, especially Figures 3, 4, and 5, and increase font sizes and adjust layouts to ensure all text and axes are clearly legible in the final version.
>
> ---
>
> Once again, we thank you for your thoughtful and constructive review. We are especially grateful for your recognition of the paper’s solidity and for highlighting its relevance to the field of scanpath modeling. We hope our responses have addressed your remaining questions and clarified our methodological choices. Your feedback helped us strengthen the manuscript further, and we look forward to incorporating these refinements in the final version.

---

### Official Review · Reviewer_3tRx · 2025-07-03

**Clarity:** 2
**Significance:** 2
**Originality:** 3
**Rating:** 4
**Confidence:** 2

**Summary:**

The authors investigate how a high-performing deep learning model (DeepGaze III) can be used to improve a more interpretable one (SceneWalk) for human gaze fixation prediction. The authors compare the predictions of DeepGaze III and SceneWalk on empirical gaze data, identify discrepancies between them, and use insights from this analysis to revise the SceneWalk model. The proposed modifications, such as incorporating saccadic momentum and time-dependent temperature scaling, lead to notable performance improvements, bringing SceneWalk closer to DeepGaze in predictive power.

**Questions:**

* How exactly is the data split for the cross-validation? Is it only split across fixations? Or are some of the stimuli and/or participants held-out as well?
* In the controversial stimuli analysis, it would help to have a quantitative analysis of the different types of prediction differences between the models.
* Overall, is there a way to make the method more systematic and less manual? The proposed method would be more impactful if it would be applicable to more models and datasets.

**Ethical Concerns:**

["NO or VERY MINOR ethics concerns only"]

**Final Justification:**

Thank you to the authors for clarifying my questions and providing more quantitative analyses. I raised my rating accordingly.

**Limitations:**

Yes

**Quality:**

2

**Strengths And Weaknesses:**

The paper presents an interesting idea: leveraging a highly-performing but difficult to interpret model to guide improvements to a more interpretable one. However, several aspects of the execution limit the paper’s impact and clarity.
* It is unclear why the use of a generative model like DeepGaze is useful to improve a more interpretable model like SceneWalk, beyond what could be achieved from the empirical data alone. The authors assert that this type of analysis is difficult with empirical data but provide no quantitative result to support this claim.
* The authors mention that the modifications to SceneWalk were already known in the literature. If these mechanisms were previously hypothesized, it is unclear why they had not been implemented in SceneWalk before. The paper would benefit from clarifying how their approach identifies these mechanisms in a more data-driven or systematic way, rather than simply reintroducing known components.
* Another unclear point is why the authors don't consider DeepGaze as a mechanistic model. While it is indeed more difficult to interpret a deep learning model, it does not necessarily mean that it doesn’t capture mechanisms of the human visual system. The main difference between DeepGaze and SceneWalk seems more to be in how easily interpretable they are.
* The authors mention that “these controversial fixations can be classified into few categories”. A quantitative analysis of these categories of prediction differences would be that claim stronger.
* The cross-validation method is crucial to evaluate the risk of overfitting in the results, but it is only briefly mentioned in the appendix and lacks detail.
* Overall, the approach presented by the authors is interesting but lacks systematicity and generality. Making the method more general and less reliant on manual analysis and insight would make it more impactful.

---

> ### Author Rebuttal · Authors · 2025-07-31
>
> We thank you for providing a thoughtful review of our work.
> We appreciate your positive assessment of the core idea and the constructive feedback: We agree that certain aspects of our methodology and contributions require clearer exposition. Your comments helped us identify clear areas for improvement in the manuscript.
>
> ### [Weakness] On the use of DeepGaze III vs analyzing empirical data directly
>
> The central difficulty lies in the structure of scanpath prediction itself. In spatial saliency modelling, it is common to compare model predictions with empirical fixation scatter plots or KDE estimates of the underlying ground truth distribution for finding the largest remaining model errors. This works well because for each image, we typically have hundreds of fixations and can form a relatively stable nonparametric estimate of the underlying spatial distribution $p(x, y)$.
>
> In scanpath prediction, the modelled distribution is more complex. Each fixation is assumed to come from a conditional distribution $p(x_i, y_i \mid  x_0, y_0, \dots, x_{i-1}, y_{i-1})$ which depends on all previous scanpath fixations. Because scanpaths are highly individual, the same fixation history is never observed more than once in practice. This means we typically have one sample per conditional distribution, making a KDE impossible. One could instead try to estimate the whole function $p(x_i, y_i \mid  x_0, y_0, \dots, x_{i-1}, y_{i-1})$ nonparametrically, e.g. with Gaussian Processes. But for typically 10 fixations in a scanpath (on MIT1003), this is a function of already 2x10=20 variables, which we would need to fit given only about ~100 datapoints (fixations) for an image. This very sparse coverage of the relevant input space is further complicated by the varying lengths of scanpath histories. In such a setting, there is no straightforward empirical baseline that allows us to say whether a given fixation is surprising or expected given prior context.
>
> Here is where DG becomes indispensable. As a high-performing, sequence-aware model trained on large-scale scanpath data, it provides a dense, queryable approximation of the true distribution $p(x_i, y_i | ...)$, allowing us to go beyond what the empirical data alone permits. For example, in Figure 2 of the paper, we highlight controversial fixations where SceneWalk performs poorly compared to DG. Without DG’s predictions as a proxy distribution, we would be left with only a single empirical datapoint to assess SW’s failure, without knowing whether that fixation was actually predictable from context.
>
> **Planned revisions:** We will update paragraphs in the introduction and Section 3 to more explicitly state this advantage, and better explain why inter-observer consistency cannot be estimated in the usual ways. We will emphasize that using a model as a "queryable stand-in" (as mentioned on lines 105-106) allows for controlled experiments on the data distribution itself, providing unique insights that guide the search for new mechanisms.
>
> ### [Weakness] On the novelty of the added mechanisms
>
> You correctly note that the effects we find (e.g., saccadic momentum) have been discussed in prior literature. However, the literature contains a myriad of reported effects that will influence fixation selection: oculomotor properties (including momentum, but also hypometric saccades, corrective saccades and saccadic adaptation), the covert attention system (pre-saccadic attention shifts, retinotopical attentional traces, spatial attention biases, attention dynamics, inhibition of return with all the different sub types and temporal dynamics, different kinds of feature attention, belief cuing,...), foveal remapping and remapping errors, different memory systems and their decay times, anatomical findings on circuit timings (e.g., cortex–colliculus loops), and cognitive influences like exploration–exploitation trade-offs, coarse-to-fine strategies or task structure.
>
> Implementing all these effects would be infeasible, even more so, because many of them are just that: effects, not yet plausibly constructed mechanisms. It is necessary to decide which of these effects might be most promising to address next. Historically, scientific intuition was used for that. Here, our approach goes one step further and provides a principled way to select those effects that seem to result in the largest prediction errors — not on datasets that have been carefully designed to maximally evoke a certain hypothesized effect, as it is often the case in the literature, but on large datasets of free-viewing behaviour on natural scenes.
>
> This is the key contribution of our work: the use of prediction disagreement on unconstrained, ecologically valid data as a systematic way to triage candidate mechanisms for inclusion in interpretable models. While the specific components we implement may not be new in themselves, our method provides a quantitative, model-based framework for assessing their individual contributions, revealing, in some cases, subtleties such as time dependencies or asymmetries in the effect shapes that had not previously been reported or appreciated.
>
> **Planned revisions:** We will revise our Methods and Introduction sections to emphasize the systematic nature of our framework. In particular, we will clarify how prediction disagreement is used to prioritize which mechanisms to implement, and how this represents a principled alternative to intuition- or experiment-driven model construction.
>
> ### [Weakness] On deep vs mechanistic models
>
> This appears to be primarily a wording issue, and we are happy to clarify our intent. Our use of the term “mechanistic model” follows a long tradition in psychology, neuroscience, and cognitive science, where “mechanistic” typically refers not just to models that implement mechanisms in some abstract sense, but to those that are explicitly constructed from interpretable components aligned with hypothesized cognitive or neural processes. However, we agree that the concept of mechanistic models in machine learning is seen as broader. We will detail the range of mechanistic models and will make clear where our work stands within that range.
>
> For a more detailed answer, please see our response to reviewer i2u5.
>
> ### [Weakness + Question] On the classification of controversial fixation
>
> For our controversial fixation analysis, we inspected a total of 72 fixations, defined as the top 6 fixations from the top 6 images in both conditions (LLD and WLLD), all shown in Appendix H. Of the four categories in Figure 2, we found:
>
> * Long saccades: 17 clear, 3 plausible
> * Short saccades: 8 clear, 1 plausible
> * Continuing: 13 clear, 2 plausible
> * Early fixations: 30 clear
>
> 66 out of 72 fixations (92%) could be clearly assigned to one of the four categories, and the remaining 6 (8%) still plausibly belonged to a known type. We see this as a strong indicator that the analysis indeed reveals failure cases that are not idiosyncratic, but rather fall into meaningful, repeatable patterns that are interpretable and mechanistically actionable.
>
> **Planned revision:** We will include a short quantitative summary in the main text (Section 5.1) and have a full breakdown and classification details in Appendix H.
>
> ### [Weakness + Question] On Cross-validation details
>
> All evaluations on MIT1003 (both for DeepGaze III and for the various SceneWalk versions) are conducted using 10-fold cross-validation *across images*. That is, images are partitioned into 10 folds, and models are always evaluated on images that were not seen during training. Both the stimuli and the associated fixations from all observers are held out during testing. This is consistent with standard practice in the saliency and scanpath prediction literature, where cross-validation across images is the norm (e.g., as done in the MIT/Tuebingen saliency benchmark), since images typically provide the strongest source of variation. Additionally, with the DAEMONS (and now Potsdam) datasets, we also test our results on completely held-out data.
>
> **Planned revisions:** We will revise the paper to make this clearer, not only in the appendix but also the main text, space permitting.
>
> ### [Weakness + Question] On “sistematicity and generality”
>
> We thank the reviewer for highlighting this important and ambitious direction. We fully agree that increasing the level of automation and generality would enhance the power and applicability of the approach.
>
> At the current stage, we intentionally chose a human-in-the-loop strategy to validate our methodology in a transparent, end-to-end fashion. Our goal was to demonstrate that targeted comparisons between a performance-driven model and a highly-interpretable one can surface actionable, theory-grounded mechanisms that measurably improve prediction. We see this as a necessary proof of concept before systematizing the pipeline further.
>
> That said, we strongly share the reviewer’s vision. Automating the identification and characterization of controversial fixations (e.g., via large vision-language models) holds clear promise. We believe approaches like VisDIFF point in the right direction, but still need to mature before fully supporting this type of model-guided scientific discovery. For now, a manual loop ensures robustness and clarity while paving the way toward a more scalable framework.
>
> **Planned revisions:** We will revise the discussion to emphasize this outlook, framing the current work as a first step toward a generalizable, semi-automated modeling pipeline, ultimately aiming for what could be seen as an “automated scientist” for behavioral modeling.
>
> ---
>
> We sincerely thank the reviewer for their thoughtful and detailed feedback. We have taken your concerns seriously and hope that our clarifications and planned revisions make the contributions of the paper clearer. If our responses addressed your doubts, we would be grateful if you considered updating your evaluation accordingly.

---

> > ### Comment · Reviewer_3tRx · 2025-08-08
> >
> > I thank the authors for their thorough and well-structured rebuttal. The rebuttal addresses several of my earlier concerns, particularly on (1) the necessity of DeepGaze, (2) cross-validation details, and (3) classification quantification. I have raised my rating accordingly. However, the work still lacks generality and validation beyond the datasets directly used to improve the model.

---

> > > ### Author Response · Authors · 2025-08-08
> > >
> > > We thank the reviewer for their reply. We are glad we could clarify several concerns and are very happy to hear that the reviewer now sees our work more positively.
> > >
> > > > However, the work still lacks generality and validation beyond the datasets directly used to improve the model.
> > >
> > > We are not sure how to understand the concern. We used only the MIT1003 dataset for our controversial fixations analysis that resulted in the implementation of the additional mechanisms in SceneWalk. To validate the usefulness of these mechanisms, we originally tested all models also on the DAEMONS dataset and now on the Potsdam Scene Corpus, thanks to another reviewer’s suggestion. Going beyond that, in the meantime we have also included COCO-Freeview in our evaluation and hence have three independent large datasets that show the benefits of our model extensions. In the table below we summarise all results for convenience. Numbers are log likelihoods (bits/fix). In parentheses, the performance explained as in Figure 6 of the manuscript:
> > >
> > > | Model                    | MIT1003       | DAEMONS       | Potsdam Corpus | COCO Freeview |
> > > |--------------------------|---------------|---------------|----------------|---------------|
> > > | Spatial baseline         | 2.516 (0%)    | 1.877 (0%)    | 1.806 (0%)     | 2.083 (0%)    |
> > > | SceneWalk (perisaccadic) | 2.659 (33.4%) | 2.716 (66.4%) | 2.321 (70.2%)  | 2.528 (56.7%) |
> > > | Scenewalk-X              | 2.821 (71.4%) | 2.889 (80.2%) | 2.426 (84.4%)  | 2.637 (70.5%) |
> > > | DeepGaze                 | 2.944 (100%)  | 3.139 (100%)  | 2.540 (100%)   | 2.868 (100%)  |
> > >
> > > We would be very grateful if the reviewer could clarify what else they expect as a test of generalization.

---

### Decision · Program_Chairs · 2025-09-17

**Decision:**

Accept (poster)

**Comment:**

This paper aims to provide a model of eye fixation that both accurately describes how humans move their eyes and also maintains interpretability, by using a deep model (DeepGaze III) to inform a mechanistic model (SceneWalk). This approach, clearly described in the paper, leads to improvements on multiple datasets, while providing mechanistic interpretability. The authors provided extensive responses to the concerns of the reviewers, who were satisfied with many of them. While the work is somewhat incremental, it still generated excitement and promises to be useful for the field. An interesting discussion occurred between a reviewer and the authors about the tradeoff between interpretability and predictive performance, which should be integrated in the paper, along with the other responses to the reviewers.